# Dynamic Linkages among Mining Production and Land Rehabilitation Efficiency in China

**Zhen Shi [1], Yingju Wu [1], Yung-ho Chiu [2,]\* Fengping Wu [1] and Changfeng Shi [1]**

[1] Business School, Hohai University, Changzhou 213022, China; 20051726@hhu.edu.cn (Z.S.); 1863510326@hhu.edu.cn (Y.W.); wfp@hhu.edu.cn (F.W.); 20161953@hhu.edu.cn (C.S.)

[2] Department of Economics, Soochow University, 56, Kueiyang St., Sec. 1, Taipei 10048, Taiwan; echiu@scu.edu.tw

**\*** Correspondence: echiu@scu.edu.tw; Tel.: +886-2-23111531 ext. 5201

**Abstract:** In the context of China's economic transformation, the consumption of mineral resources plays an important role in its economy's sustainable development, and so improving mining efficiency is regarded as the basis of industrial development. However, in the pursuit of mine exploitation, the destruction of land resources has attracted greater attention by government and society, with many scholars focusing more on land rehabilitation in recent years. Thus, from the perspective of climate change, this research synthetically analyzes the two stages of mining production and land rehabilitation, by applying mining employees, fixed assets' investment stock, production of non-petroleum mineral resources, accumulated destruction of land area, rehabilitation investment, rehabilitation of land area, and average temperature to the dynamic two-stage directional-distance-function data envelopment analysis (DEA) model under exogenous variables for 29 provinces in China. The results show that the overall efficiency of mining-production-land rehabilitation in most provinces fluctuates around 0.5 and spans a large range of improvement. The efficiency of the mining production stage fluctuates around 0.55 and is relatively flat over four years. The efficiency of the land rehabilitation stage fluctuates during the four years, with it being higher in 2014, but lower in 2015. Generally speaking, the efficiency of the land rehabilitation stage is higher, promoting the improvement of overall efficiency, but the efficiencies of some provinces' land rehabilitation stage are quite different, as some provinces still need to improve their overall efficiency level. There are also differences in the efficiencies of each decision-making units (DMU)'s variables. In sum, China should initiate corresponding policies according to specific situations, promote scientific mining in each province, and coordinate the development of mining production and land rehabilitation.

**Keywords:** mining production; land rehabilitation efficiency; two-stage dynamic DEA; meta-frontier; undesirable outputs

## 1. Introduction

Mining plays an important role in promoting China's economic transformation and sustainable and efficient development. With the advancement of reforms and opening up, the country's gross national product (GDP) has maintained rapid growth. Its proven mineral resources account for about 12% of the world's total, ranking third, and 158 kinds of minerals have been found throughout the country. According to its proven reserves, 25 of China's 45 major minerals rank in the top three globally, of which 12 are rare earth, gypsum, vanadium, titanium, tantalum, tungsten, bentonite, graphite, Glauber's salt, barite, magnesite, and antimony. China is certainly one of the few countries in the world with abundant mineral resources, a broad range of mineral resources, and supportive

service facilities around mining areas, such as water (water source), electricity, roads, etc. As an important non-petroleum mineral resource, coal made up around 60% of mining in 2017, and although its proportion within total energy consumption has gradually fallen, it still plays an important role in China's energy supply. At present, over 95% of energy, over 80% of industrial raw materials, and over 70% of the agricultural means of production come from mineral resources. This shows how mining and the development of mines provide the necessary material basis and resource guarantee for the development of China's economy and help improve the employment level and income level of its citizens. Therefore, its government should pay particular attention to the improvement of mining input–output efficiency.

The process of mining not only leads to the destruction of land, but also to the damaging of property and loss of life [1]. In China's pursuit of mining production, it must consider the rehabilitation of destructed land caused by mining. For example, for the period 2012–2017, the accumulated destructed or occupied land totaled over 2 million hectares. In 2013, Inner Mongolia saw 50 million hectares of land occupied or destroyed during the mining process, which is about 100 times higher than that in 2012. In 2013, the accumulated destructed or occupied land in all provinces of China increased by about 17 times, compared with that in 2012, while non-petroleum mineral resources' production in the same year showed a downward trend. This shows that the contradiction between mineral exploitation and land destruction caused by mining is becoming increasingly fiercer.

Starting with the outline of the 13th five-year plan, China's economic path has changed from high-speed development to high-quality development, among which investment in the mining industry continues to decline, while land protection requirements for resource exploitation are increasing. In 2016–2017, the area of rehabilitation due to mining in China increased by over 80%. Driven by government policies, land destructed or occupied by mining in China is being alleviated. President Xi Jinping pointed out at the 19th Communist Party of China (CPC) National Congress, on October 18, 2017, that the country should target harmonious coexistence between mankind and nature and practice the concept of "Clear waters and green mountains are as good as mountains of gold and silver". On August 31, 2018, the 5th Session of the 13th National People's Congress (NPC) passed a law for the People's Republic of China, on the prevention and control of soil pollution, which regulates prevention and protection, risk control and repair, and guarantee and supervision. Among them, risk control and restoration are distinguishable between agricultural land and construction land. On April 27, 2019, the Office of the Work Safety Committee of the State Council issued a notice on the closure of non-coal mines that do not meet specific conditions for safe production. The notice pointed out that the task of ensuring the closure of over 1000 non-coal-mine mountains (including tailings ponds) that do not meet such requirements is to be completed in 2019, so as to prevent and defuse major safety risks at the source. For land-destruction problems caused by mining, the government uses administrative means to further strengthen land protection. When insisting upon sustainable development, China should focus on both the economic output brought by mining and the land rehabilitation after mining. Problems and solutions to this process have led to extensive investigation by Chinese scholars.

As the material basis for the survival of a country, land resources are also affected by natural factors, such as changes in weather and climate patterns. In northern mines, climate factors have an important impact on open-pit mining and land restoration—for example, (1) the difficulty of mining equipment maintenance increases in winter; (2) in freezing winter, permafrost and other factors can negatively impact mining; and (3) during mining restoration, a cold climate aggravates the process, while high temperatures in summer affect and interfere with vegetation restoration in the process of land restoration. Therefore, when exploring the overall efficiency of mining production and land rehabilitation, climate factors, as external variables, become interference factors to mining and land rehabilitation.

To fully understand the problems existing in the mining process in China, one must explore the amount of mineral resources and the degree of land destruction during the process, as well as the efficiency of land rehabilitation. By finding the weak link of low efficiency, provinces in China can

then implement practical policies by using existing resources that promote a balance between mining and land protection. Data envelopment analysis (DEA) is a relatively effective method for evaluating the efficiency of decision-making units (DMU) and has been widely used in different industries and departments, as it has a stronger advantage in dealing with multi-index inputs and multi-index outputs, making it a favorite of many scholars around world and suitable for this research topic.

Most studies in the existing literature on industrial environmental pollution are from the perspective of macro-environmental pollution, or they separately analyze coal among mining resources. In general, scholars have used one-stage DEA to look into the damage to land caused by mining. Based on the dynamic two-stage directional distance function DEA model under the exogenous variables model, this paper studies 29 provinces of China (not including Tibet and Shanghai, due to a lack of data, and excluding Hong Kong, Macao, and Taiwan). This paper calculates the overall efficiencies of mining production and land rehabilitation and the efficiency of each stage for 2014–2017 and also studies the efficiency of each variable on mining production and land rehabilitation, so as to put forward specific suggestions for each province's own situation that can improve mining production and land destruction efficiencies.

This paper's contribution to the development and improvement of China's mining areas mainly includes the following points. First, it is a pioneer research that explores mining production and land destruction in 29 provinces of China, from the national level. In the mining production stage, fixed assets' investment stock and mining employees are regarded as the input variables, non-petroleum mineral resources are the desirable output, and accumulated destruction of land area is the undesirable output. On this basis, the accumulated destruction of land area is the intermediate variable of the two stages. In the land rehabilitation stage, rehabilitation investment is the input variable, and rehabilitation of land area is the output variable. Second, by comparing mining production and land rehabilitation efficiency of 29 provinces in China, this paper observes the overall efficiency for each year and each stage, demonstrating the contribution and misappropriation of various variables in each region to total efficiency. Third, we discuss the characteristics and numerical fluctuations of the mining production stage and land rehabilitation stage from a local perspective and put forward specific suggestions for the efficiency changes of each province, providing guiding opinions for the sustainable development of China's mining industry. Fourth, this paper introduces local annual temperature as an exogenous variable, and by adding this, the influence of temperature changes in the mining area can be taken into account and therefore help provide a reasonable conclusion for the evaluation of a mining area.

## 2. Literature Review

Environmental pollution caused by mining is not a unique problem in China. Using remote-sensing-GIS (geographic information system) techniques, it was found that the Jharia coalfield in India also resulted in the reduction of vegetation covering [2]. Ghana in Africa has mapped land-cover images of gold-mining areas in Western Ghana through satellite images that show open-pit mining there has resulted in 58% deforestation and 45% loss of farmland [3]. Effective improvement and solution to the problem of land damage in the mining process are not only of great significance to the high-quality development of China, but are also relevant to the mining and management of all regions around the world. It can be seen in its industrial transformation that China urgently needs to accelerate the transformation of its mining operation mode and mining volume and pay attention to both mining efficiency and environmental protection.

The majority of scholarly research on mining and land destruction explores the topic from the following three aspects: mining production and environmental pollution; land pollution control; and unified management of production and land destruction. Below, we present them in greater detail.

### 2.1. Research on Mining Production and Land Pollution

Venkateswarlu, Nirola et al. used mineralogical analysis and physiochemical analysis on abandoned metalliferous mines (AMLs), showing that a lack of awareness for timely management of the environment surrounding a metal mine site results in several adverse consequences, such as

rampant business losses, abandonment of the highly lucrative mining industry, domestic instability and a rise in ghost towns, increased environmental pollution, and indirect long-term impacts on the ecosystem [4]. In view of the situation at that time, some suggestions on metal mine reclamation were put forward, such as enhancing the stability of native plants through microbially enhanced phytoremediation and nanotechnology for the efficient reclamation of AMLs and initiating laws governing the management of mined sites and ecological restoration, so as to effectively promote the improvement of AMLs' environment. Bhuiyan et al. used multivariate statistical analysis, principal component analysis, and cluster analysis to evaluate heavy metals in mine drainage soil and surrounding farmland in Northern Bangladesh. They employed energy dispersive X-ray fluorescence (EDXRF) methods to carry out the mining soil samples and applied enrichment factors (EF), geo-accumulation index ($I_{geo}$), pollution load index (PLI), etc. to evaluate the situation of soil pollution. Their results showed that the average concentrations of titanium (Ti), manganese (Mn), zinc (Zn), lead (Pb), arsenic (As), ferrum (Fe), rubidium (Rb), strontium (Sr), niobium (Nb), and Zirconium (Zr) in the soil of this area exceed the world's normal average values, and in some cases, Mn, Zn, and Pb surpass the toxicity limits of corresponding metals. PLI (pollution impact index) shows that pollution in the far part of the affected area is the most serious (PLI of 4.02) [5].

Li et al. utilized multiple linear regression models to analyze and study China's coal resources from 1997 to 2010. They took the variables raw coal (RCO), coal industrial output (GCIOV), new investment in fixed assets of coal (CFANI), gross domestic product (GDP), and gross value of industrial output (GIOV) to measure the relationship between coal industry development and economic growth. They showed that environmental degradation caused by coal mining and washing is calculated by the given environmental damage cost model, and that there is a significantly positive correlation between coal development and economic growth, in which the total environmental loss of coal mining and washing covers approximately 2.7% of the average price of coal. This method uses a multiple regression to investigate the linear relationship between the variables of coal mining on economic development and environmental damage, but it does not effectively consider the time factor in the model, as well as the impact of individual variables on the environment and economy [6].

Razo et al. employed a pH measurement, X-ray diffraction analysis, principal component analysis (PCA), and contour mapping, to test the area of Villa de la Paz-Matehuala, San Luis Potosi (Mexico). Their results showed that cuprum (Cu) in the soil samples of the mining site is seriously over regulatory standards, and the maximum arsenic concentration in the local water tank exceeds that in Mexico by five times the drinking-water-quality guidelines [7]. Du et al. applied the coefficients of heavy metal enrichments (CHMEs) to evaluate the capability of crop accumulating heavy metals and the availability and mobility of heavy metals in soils and used target hazard quotients (THQs) to assess health risks from heavy metals in Hunan, China. The results showed that the average value of Cadmium (Cd) in the soil is 0.13–6.02 mg kg$^{-1}$, the over standard rate is 59.6%, and total-Cadmium (T-Cd) in the soil varies greatly, with the coefficient of variation reaching 146.4%. The regression results also denoted that a significant correlation exists between T-Cd and hydrochloric acid-Cd (HCI-Cd) in the soil (r = 0.77, $\varrho$ < 0.01) [8]. Li et al. used cold vapor atomic absorption spectrometry (CVAAS) and a dual-stage Au amalgamation method to calculate the total Hg (T-Hg) content in wastewater, steam water, soil, and moss samples in the mining area of Tongren, Guizhou, and determined that pollution to the local environment is caused by manual mercury mining. Thus, it indicates that the mining of mercury causes heavy metal pollution to the paddy fields due to atmospheric subsidence and mercury settling to the soil surface in the local mining area of Tongren [9].

Li et al. used the geo accumulation index and data from the U.S. Environmental Protection Agency (USEPA), to evaluate heavy metal polluted soils originating from mining areas in China. By quantifying the harm of soil pollution to human health in a mining area, their results showed that heavy-metal pollution not only causes harm to the environment, but also brings high cancer risk to people [10]. Acosta et al. utilized multivariable statistical and spatial analyses to calculate two tailing ponds (Lirio and Gorguel) from an abandoned Pb-Zn mine. The soil samples were tested by using

the pH value and electrical conductivity (EC). The article also used the correlation matrix and PCA to evaluate heavy-metal-waste properties, showing results that the two ponds are polluted by Cd, Pb, and Zn. In the flow of the natural environment, polluted soil will even infiltrate into groundwater, causing serious environmental pollution [11]. Simon et al. utilized the structural development index (SDI) and inductively coupled plasma mass spectrometry (ICP-MS) to study pyrite in Southern Spain, with results showing that the main pollutants in the mining area are Zn, Pb, Cu, As, Sb, Bi, CD, and Tl. The permeability of tailings depends on the soil properties, in which Zn exceeds the maximum concentration range allowed by the international community. However, due to a drier climate, the drying of tailings and the accompanying aeration make sulfide become oxidized to sulfate, thus alleviating pollution. If the tailings are not cleaned up quickly, then future rainfall may aggravate the pollution problem. This article pointed out that environmental factors also make a difference on mining production [12].

Wang et al. discussed the degree of pollution and spatial distribution role of heavy metals in agricultural soil of a Sb mining area and determined the heavy-metals content and metal concentration of 29 environmental samples collected by using aqua regia, ICP-MS, atomic fluorescence spectrometry (AFS-2202), and inductively coupled plasma atomic emission spectrometry (ICP-AES). The results showed that Sb is over the standard, and the soil is polluted seriously by heavy metals. The heavy-metal concentrations in the topsoil of the Xikuangshan area are mostly higher than the background values, especially Sb and As. Heavy metal pollution is caused by both mining activities and agricultural activities [13]. Equeenuddin et al. explored and studied coal mining in Northeast India. The main environmental pollution problem studied is acid mine drainage (AMD), and the nature of direct mine discharge is high acidity to alkalinity (up to pH 7.6). The results showed that Mn, Fe, and Pb are seriously polluted in groundwater resources of the mining area, while the groundwaters close to the collieries and AMD affected creeks are highly polluted, but the major rivers are less polluted by AMD. It can be seen that research on mining and land destruction in the current literature is mostly concentrated on the mining and destruction of individual minerals, with a lack of research on China's overall mining production [14].

## 2.2. Research on Mining Pollution Treatment

Andres and Mateos used de-trended correspondence analysis (DCA) and canonical correspondence analysis (CCA) to evaluate the mining rehabilitation situation of a mining area in Santa Margarida, Spain. The article primarily evaluated four post-mining-restoration treatments (soil spreading, soil spreading + grass and herb sowing, soil spreading + tree planting, and soil spreading + sowing + planting). However, after 12 years of restoration, no one treatment method can achieve the restoration of forest soil pretreatment conditions, with soil spreading as the most ineffective treatment method, and soil spreading + sowing and soil spreading + sowing + planting inducing grassland soil conditions. In their research, the number of taxa, diversity of species, and communities of Collembola and Orion were used to evaluate the effect of soil restoration [15]. Hilson commented on current research and policies in dealing with mercury pollution in gold mines. The results showed that the technology implemented to improve any harm to workers and operators on acute mercury exposure and to prevent further pollution has little effect on mercury pollution control in mining areas. Only by increasing the understanding of small-scale gold mining communities and enhancing the acquisition of knowledge about mercury pollution by regional governments and donor agencies can environmental solutions and appropriate policy support be promoted in mining areas [16]. Macias et al. discussed the metal removal process of AMD by a pilot multistep passive remediation system. The reactive transport model predicts that 1 m$^3$ of MgO-DAS can treat a flow of 0.5L/min of highly acidic water (total acidity of 788 mg/L CaCO$_3$) for more than three years [17].

By analyzing the distribution of resources in China's metal mines, Chen put forward two methods for pollution control in its mines. The first is to treat and manage the pollution of tailings from the source, so as to ensure the deep treatment of mineral water in the beneficiation process. The second is to repair the pollution situation from physicochemical and phytoremediation technologies, so as to realize sustainable development of the mining area [18]. Akcil and Koldas introduced the

generation and processing technology of AMD, and their document divided mine pollution control into three levels. The first level is primary prevention, which is control of the acid production process. The second level is secondary control, which is the deployment of acid discharge prevention measures. The third level is tertiary control, which is the collection and treatment of wastewater [19]. Zhang et al. utilized the life-cycle assessment (LCA) method to study an opencast coal mining area in Yimin. Their research took 100 tons of coal production as the functional unit, evaluating the environmental risks of coal stripping, mining, transportation, processing, and reclamation. Through the assessment of environmental risk in the production stage, the results showed that the contribution rate of dust to the environmental impact is 36.81%, which is the largest. In order to slow down the pollution of a mining area, the paper put forward some suggestions, such as sprinkling water, clean transportation, improving processing efficiency, etc. At present, most of the literature available for reference is based on presenting relevant governance methods on specific mining areas and for macro-size non-petroleum mineral areas, but there are no relevant governance and solutions [20].

Grant used a state-and-transition successional model to assist Alcoa in determining which locations will or will not meet the established restored completion criteria. Five desirable and nine deviated states were identified and described in detail. The results showed in the 6429 hectares of local species recovered between 1991 and 2002 that 98% of the species are at or above the desired successional trajectory [21]. Based on the analysis of the main environmental problems in the mining area, Lei et al. put forward the application of the landscape strategy and "natural" technology for mine ecological restoration. In addition, a multi-objective integration method based on landscape planning was proposed to protect and reasonably utilize the mining land through local or regional actions and cooperation, so as to realize the sustainable development of China's mining industry [22]. Neri and Sanchez tested the nine limestone quarries and conformity indices, showing results that most of the quarry planning practice is poor, 50% of quarry businesses hit a high level of compliance, and management practice has achieved moderate conformity [23]. Based on the background of Australia's post-mined landscape, Doley et al. proposed a framework with the special features of "novel" ecosystems and agro-ecosystems having a range of business and social values that can bridge the conceptual gap and that separate the ecological functions of under-restored (e.g., derelict sites) from re-instated "natural" landscapes [24]. Anawar et al. conducted an analysis on vegetation development and community succession in the abandoned Sao Domingos pyrite mining area. Their conclusion is that appropriate soil development that is rich in nutrients and nature of mining waste is conducive to promoting high vegetation growth, mine restoration, and ecological restoration of degraded land in mining areas [25].

## 2.3. Research on Mining Production and Pollution Treatment

Song et al. established the non-radial, non-angular slacks-based measurement (SBM) model to comprehensively evaluate the operating efficiency of 36 coal companies in China. The DEA model is also used to explore the efficiency of the production and pollution control process of coal companies. In the production process, non-current and current assets are selected as input variables in the production process, while the desired variables in the output stage are set as the operating income. Considering the unexpected output in asset flow and equipment operation, they are selected as alternative values. Based on the input redundancy and output deficiency of each enterprise, their article coordinated and unified the production and environmental management of coal enterprises and put forward suggestions for the production of coal enterprises [26]. Liu et al. relied on a range-adjusted measure model of DEA and built three projects, including operational efficiency, environmental efficiency, and unified efficiency, to reflect the sustainability impact of this integration policy. That study compared the two representative provinces of Shanxi and Inner Mongolia (2005–2012), evaluating the policy effect of the transformation of coal industry to sustainable development. The results suggested that, although the integration reform of China's coal industry has a negative impact on coal production capacity, it has significantly improved the industry's environmental performance. However, the method of nationalization cannot exceed that of market orientation, because the inefficiency of capital factors offsets the results of efficiency improvement [27].

Wang et al. used an index system evaluation method to evaluate the diversity of Mineral Resources Carrying Capacity (MRCC) in China's mining cities. Their research divided the city development potential into one level and put forward a guarantee period. The development potential of regional mineral resources is evaluated by calculating the guarantee period of each region. Their results showed that MRCC has a strong development potential in the mining cities of central and western regions and for developing mining cities. Mature and developing mining cities have strong development potential, while the environmental pollution of developing mining cities is more serious. The coordination between the development of mineral resources and the environmental protection of western and developing mining cities needs to be improved [28]. Li et al. established the modified dynamic DEA SBM model, with the intermediate variable set as fixed assets. The DEA SBM model is used to analyze the relationship between coal production and land destruction. The results showed that the coal-production efficiencies of Shanxi, Inner Mongolia, Fujian, Jiangxi, Shandong, Guizhou, and Shaanxi are not very obvious, the land-loss efficiencies of 17 provinces vary quite a bit, and 15 provinces need to be improved. Therefore, it is necessary to speed up the exploitation of new energy resources for coal mining and to reduce land damage [29]. Based on the Range-Adjusted Measure (RAM) method, Wang et al. analyzed China's regional energy and environmental efficiency in 2006–2010. By introducing the economic concepts of natural disposability and managerial disposability, the method replaces strong and weak disposability in conventional environmental efficiency evaluation. The results showed that Beijing, Shanghai, and Guangdong have the highest integrated energy and environmental efficiency during the study period, which can be used as a benchmark for other inefficient areas. Moreover, China's average production efficiency has slightly decreased, while the average emission efficiency has slightly increased and still has great potential for energy conservation and emission reduction. Therefore, the document suggested that China should pay more attention to technological innovation, to improve energy efficiency [30].

## 3. Model and Method

The DEA model can be divided into radial DEA model and non-radial DEA model. In the case of Charnes et al. with fixed returns to scale, the CCR model is proposed, and Banker et al. then replaced the constant returns to scale (CRS) BCC model with variable returns to scale (VRS) [31,32]. The non-radial DEA model is represented by the slack-based measure (SBM) model proposed by Tone [33]. In addition to the CCR, BCC, and SBM models, Chung et al. set up the directional distance function ray DEA model, with the direction vector composed of input direction vector and output direction vector. The direction vector values of different input and output indicators represent their degrees of relative priority (or importance). This model can also be applied to undesirable output and has been applied to the fields of energy, environment, and ecology. However, Chung et al. proposed the concept of the radial distribution function (RDF), which is an extended RDF [34]. The traditional directional distance function (DDF) is a ray measurement model, but efficiency calculation fails to cover all non-zero margins, meaning all inefficient sources are covered. Therefore, the efficiency value is overestimated, and non-radial DDF has the advantage of being a pragmatic measure of manufacturers' operational efficiency. In order to solve this kind of problem, Fare and Grosskopf established a non-directed distance function that is better than other methods, because it provides a more reasonable and accurate estimate [35].

When traditional DEA is evaluated for efficiency, it transforms the efficiency between the two variables through input and output projects, and the conversion process is identified as a "black box". Fare et al. considered that the production process is composed of multiple production technologies, and a sub-production technology is regarded as a sub-DMU, and hence the efficiency of each sub-process can be analyzed [36]. Chen and Zhu, Kao and Hwang, and Kao linked all stages through some intermediate outputs [37–39]. By linking the stages in this way, they calculated the efficiencies of each stage under different conditions. After Fare et al., Tone and Tsutsui put forward the weighted slack-based measures model [40]. The analysis basis of the network DEA model takes the linkage among departments of decision-making units and regards each department as a sub-DMU, to find the optimal solution. In the network DEA model, the dynamic method is also allowed, in which

DMUs are evaluated at different periods of time, and a carryover is introduced to connect different stages of DMUs in different periods [41]. In the development of dynamic DEA, Klopp proposed window analysis and first used dynamic analysis [42]. Fare and Grosskopf were the first to put interconnecting activities into the dynamics, while Fare and Grosskopf presented dynamic DEA modification and extension [43,44]. Following Fare et al., Tone and Tsutsui extended the model to a dynamic analysis of a slacks-based measure [36,41]. Tone and Tsutsui once again proposed the weighted SBM dynamic network DEA model, taking the linkage among departments of decision-making units as the analysis basis of the network DEA model and regarded each department as a sub-DMU, with carryover activities as the linkage [45].

In this paper, we consider the influence of exogenous variables of climate change on the efficiency of mining and land destruction. Therefore, we refer to the concept of a dynamic two-stage model and add the DDF model, to consider the exogenous variables for climate change. Thus, to solve the shortage of static one-stage and exogenous variables, this paper proposes the dynamic two-stage directional distance function DEA, model under exogenous variables. At the same time, to fully understand the overall picture of industrial operation performance and avoid the underestimation or overestimation of efficiency value and improvement space, an empirical study of the new model considers the impact of average temperature change on mining production and land destruction efficiency. Below, we propose the dynamic two-stage DDF, considering exogenous variables for climate change.

### 3.1. Dynamic Two-Stage DDF with Exogenous Variables

Suppose DMU has two stages (mining production stage and land rehabilitation stage) in each time period t $(t = 1, \ldots, T)$. The mining production stage has M inputs $X_{ij}^t (i = 1, \cdots m)$, D intermediate products $Z_{dj}^t (d = 1, \cdots D)$, and K desirable outputs $q_{kj}^t (k = 1, \cdots K)$. The land rehabilitation stage includes G inputs $W_{gj}^t (d = 1, \cdots G)$ and S desirable outputs $y_{rj}^t (r = 1, \cdots S)$.

$Z_{dj}^t (d = 1, \cdots D)$: links the mining production stage and land rehabilitation stage;

$C_{hj}^{t-1} (h = 1, \cdots H)$: the carryover variable;

V: external variables;

$X_{ij}^t$: the inputs of mining production are mining employees

$Z_{dj}^t$: the link between the mining production stage and land rehabilitation stage, which is accumulated destruction land area;

$q_{kj}^t$: the output of mining production, which is production of non-petroleum mineral resources;

$W_{gj}^t$: the input of land rehabilitation stage is rehabilitation investment;

$y_{rj}^t$: rehabilitation of land area;

$C_{hj}^{t-1}$: fixed assets investment stock;

$b_{uj}^t (U = 1, \cdots V)$: average temperature

The model is as follows:

$$\text{Max } \sum_{t=1}^{T} V_t (W_1^t \theta_1^t + W_2^t \theta_2^t) \tag{1}$$

S.T. is the constraints:

| Mining production stage | Land rehabilitation stage |
|---|---|

$$\sum_{j}^{n} \lambda_j^t X_{ij}^t \leq \theta_1^t X_{ip}^t \ \forall i \ \forall t \qquad\qquad \sum_{j}^{n} \mu_j^t Z_{dj}^t \leq \theta_2^t Z_d^t \ \forall d \ \forall t$$

$$\sum_{j}^{n} \lambda_j^t z_{dj}^t \leq \theta_1^t z_{dp}^t \ \forall d \ \forall t \qquad\qquad \sum_{j}^{n} \mu_j^t Z_{dj}^t \leq \theta_2^t Z_d^t \ \forall d \ \forall t$$

$$\sum_{j}^{n} \lambda_j^t q_{kj}^t \geq \theta_1^t q_k^t \ \forall k \ \forall t \qquad\qquad \sum_{j}^{n} \mu_j^t w_{gj}^t \leq \theta_2^t w_g^t \ \forall g \ \forall t$$

$$\sum_{j}^{n} \lambda_j^k \leq \ \forall t \qquad\qquad\qquad \sum_{j}^{n} \mu_j^t = 1 \ \forall t$$

$$\lambda_j^t \geq 0 \ \forall j \ \forall t \qquad\qquad\qquad\qquad \mu_j^t \geq 0 \ \forall j \ \forall t$$

The exogenous variables are as follows:

$$\sum_{j=1}^T \lambda_1^t b_{Uj}^t = \theta_1^t b_U^t \quad \forall U \ \forall t \tag{2}$$

The link between the two stages is as follows:

$$\sum_{j=1}^n \lambda_j^t Z_{dj}^t = \sum_{j=1}^n \mu_j^t Z_{dj}^t \quad \forall d \ \forall t \tag{3}$$

The link between the two periods is as follows:

$$\sum_{j=1}^n \lambda_j^{t-1} c_{hj}^t = \sum_{j=1}^n \lambda_j^t c_{hj}^t \quad \forall h \ \forall t \tag{4}$$

Among them, $\gamma_t$ is the weight assigned to time period, $t$; and $w_1^t$ and $w_2^t$ are the weights assigned to the mining production stage and land rehabilitation stage in time period, $t$, respectively. Therefore, for each time period ($t$), $w_1^t, w_2^t, \gamma_t \geq 1$, and $\sum_{t=1}^T \gamma_t = 1$.

We can calculate the following four efficiency groups through the linear programming formula in Equations (5–8).

The efficiency of the mining production stage is as follows:

$$\rho_1^t = 1 - \theta_l^{t^*}; \text{l=1,2;} \quad t = 1,2,\cdots,T \tag{5}$$

The efficiency of the land rehabilitation stage is as follows:

$$\rho_2^t = 1 - \sum_{t=1}^T \gamma_t \theta_t^{t^*}; \text{l=1,2} \tag{6}$$

The period efficiency value is as follows:

To evaluate the overall efficiency of each period, $T$, of the DMU being evaluated in this group, we express it as follows:

$$\rho^t = w_1^t \rho_1^t + w_2^t \rho_2^t; \ t = 1,2,\cdots,T \tag{7}$$

The DMU's overall efficiency is evaluated in this group and is given by the weighted sum of periodic efficiency on t, which can be expressed as:

$$\rho = \sum_{t=1}^T \gamma_t \rho^t \tag{8}$$

### 3.2. Input, Desirable Output, and Undesirable Output Efficiency

We use Hu and Wang's total-factor energy-efficiency index to overcome any possible biases in the traditional energy efficiency indicators, for which there are four key efficiency models: production of non-petroleum mineral resources, accumulated destruction of land area, rehabilitation investment, and rehabilitation of land area [46]. "$I$" represents area, and "$t$" represents time. The efficiency models are defined in the following:

$$Input\ efficiency = \frac{Target\ input}{Actual\ input} \tag{9}$$

$$Undesirable\ output\ efficiency = \frac{Target\ Undesirable\ output}{Actual\ Undesirable\ output} \tag{10}$$

$$Desirable\ output\ efficiency = \frac{Target\ Desirable\ output}{Actual\ Desirable\ output} \tag{11}$$

If the target inputs equal the actual inputs, then the efficiencies are 1 and indicate overall efficiency; however, if the target inputs are less than the actual inputs, then the efficiencies are less than 1 and indicate overall inefficiency.

If the target desirable outputs are equal to the actual desirable outputs, then the efficiencies are 1 and indicate overall efficiency; however, if the target desirable outputs are more than the actual desirable outputs, then the efficiencies are less than 1 and indicate overall inefficiency.

If the target undesirable outputs are equal to the actual undesirable outputs, then the efficiencies are 1 and indicate overall efficiency; however, if the target undesirable outputs are less than the actual undesirable outputs, then the efficiencies are less than 1 and indicate overall inefficiency.

## 4. Data, Index, and Empirical Analyses

In this section, we first describe the study area and the meaning of the variables and apply the dynamic two-stage directional distance function DEA model under exogenous variables model to calculate mining production stage and land rehabilitation stage efficiencies, overall efficiency, and variable efficiency. We also carry out an empirical analysis on the provinces.

### 4.1. Data and Variables

This study uses panel data from 2014 to 2017 for 29 provinces in China. We use the variables in this paper to interpret and statistically analyze the data.

#### 4.1.1. Explanation of Variables

This paper takes 29 provinces (cities and districts) in China as the subjects of research. Therefore, the research areas of this literature are Beijing, Tianjin, Chongqing, Hebei, Shanxi, Liaoning, Jilin, Heilongjiang, Jiangsu, Zhejiang, Anhui, Fujian, Jiangxi, Shandong, Henan, Hubei, Hunan, Guangdong, Hainan, Sichuan, Guizhou, Yunnan, Shaanxi, Gansu, Qinghai, Inner Mongolia, Guangxi, Ningxia, and Xinjiang.

In the mining production stage, mining employee and fixed assets' investment stock are used as input variables. One output variable, accumulated destruction of land area, is an undesired output. Because oil and gas mining mostly entails deep mining and has little impact on the surface, it is not included in the scope of variable statistics. The desirable output is set as production of non-petroleum mineral resources. In the land rehabilitation stage, this paper takes rehabilitation investment as the input variable and rehabilitation of land area as the output variable. The accuracy of the two-stage data is increased by introducing average temperature as the climate exogenous variable. Accumulated destruction of land area is selected as an intermediate factor to connect the mining production stage and land rehabilitation stage. Table 1 shows the data for the specific variables.

**Table 1.** Input and output variables.

| Stage | | Variable | Unit |
|---|---|---|---|
| Stage 1 | Input | Mining Employees | 10,000 people |
| | | Fixed Assets' Investment Stock | 100 million RMB |
| | Output | Production of Non-Petroleum Mineral Resources | 10,000 tons |
| | | Accumulated Destruction of Land Area | Hectare |
| Stage 2 | Input | Rehabilitation Investment | 100 million RMB |
| | Output | Rehabilitation of Land Area | Hectare |
| Climate variable | | Average Temperature | Centigrade |

Figure 1 shows the flow structure of this paper by using flow chart. See Figure 1 for details.

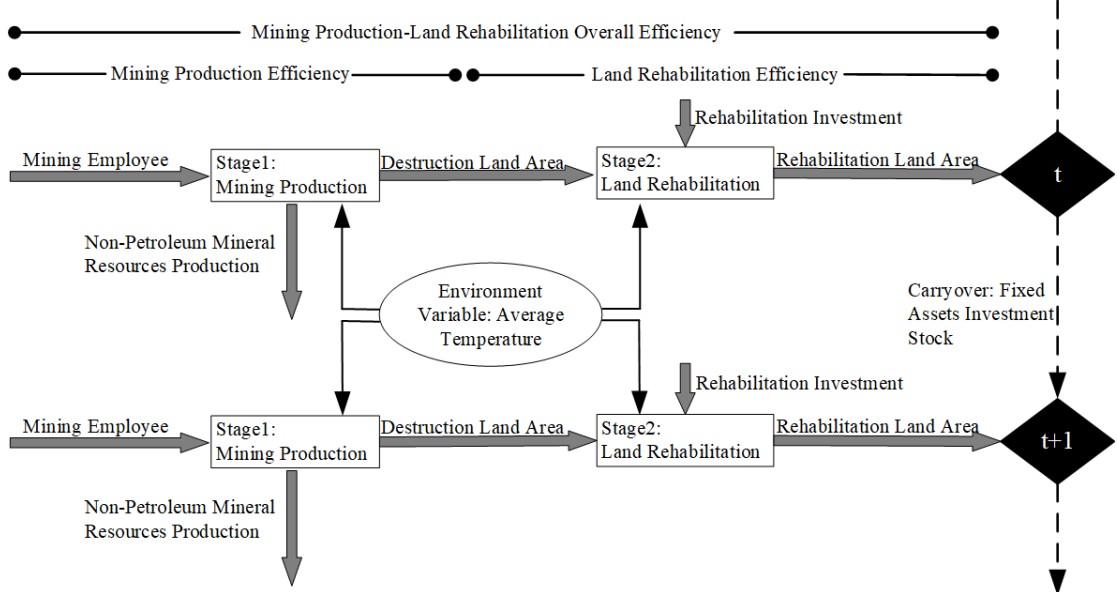

**Figure 1.** Network dynamic model.

The raw data for the mining employees and fixed assets' investment stock are from the China Statistical Yearbook 2014–2018. The raw data for the production of non-petroleum mineral resources are from the China Land and Resources Statistical Yearbook 2014–2018. The raw data for rehabilitation investment, rehabilitation of land area, and accumulated destruction of land area are from the China Statistical Yearbook on Environment 2014–2018. The raw data for the average temperature are from the China Meteorological Yearbook. Some data on rehabilitation investment and rehabilitation of land area are missing, and the missing data are calculated by the interpolation method. In some provinces, the average temperature is missing and replaced by the average temperature of the provincial capital.

The specific variables are explained below.

①Average Temperature (AT): It is the sum of the monthly average temperatures of the 12 months in the year divided by 12 or the annual average temperature. The average annual temperature in some of these provinces is in the range of intervals; herein, the maximum value and the minimum value are accumulated to get the average value.

②Mining Employees (ME): This number of employees worked in mining industry on the last day of the reporting period and received wages or other forms of remuneration for their work.

③Fixed Assets' Investment Stock (IS): This paper considers the "perpetual inventory method" of depreciation of capital stock. According to the depreciation of fixed assets in China, the depreciation rate of material capital is set as 0.096. The formula for the perpetual inventory method is as follows:

$$K_{it} = K_{i,t-1}(1 - \delta) + I_{it} \tag{12}$$

where $K_{it}$ and $K_{i,t-1}$ are the investment stock of the current year and the investment stock of the previous year, respectively, and δ represents the depreciation rate.

④Production of Non-Petroleum Mineral Resources (MP): For the exploitation of non-petroleum mineral resources by various regions, non-petroleum resources refer to the non-oil and gas state and solid natural enrichment produced by geological action, most of which are contained within the surface or crust of the land.

⑤Accumulated Destruction of Land Area (AD): The total land area occupied or destructed by tailings, discharged solid waste, open-pit mining, mining collapses, and other geological disasters in mines at the end of the reporting period.

⑥Rehabilitation Investment (RI): It refers to investment for the rehabilitation of the mine environment during the reporting period, including central finance, local finance and mining enterprise input, and private investment.

⑦Rehabilitation of Land Area (RA): It refers to the area of rehabilitation of governance during the reporting period, including reclamation, ground-collapse management, forest, grass, construction, use, and so on.

### 4.1.2. Data Description

The average temperature in various regions of China is influenced by latitude factors. The maximum temperature is 25.30 ℃, while the minimum temperature is only 2.90 ℃, and so there is differentiated regional distribution. However, due to the particularity of average temperature factors, the standard deviation is 5.44, and the temperature horizontal differences between regions are low. The maximum number of mining employees is 0.99 million. In this variable, the standard deviation is low, at only is 20.35. The average production of non-petroleum mineral resources is 275.76 million tons, the maximum value is 962.93 million tons, and the minimum value is 9.32 million tons, greatly correlating to the distribution of natural resources in various regions. The maximum value of accumulated destruction land area is 926,606 hectares, and the minimum value is 1646 hectares. The average amount of rehabilitation investment is 4,097,063 million RMB, and the sample standard deviation is 46,298.26, closely correlating to the accumulated destruction of land area in various regions. The average rehabilitation of land area is 1755.47 hectares, while the minimum value is only 22 hectares, and the maximum value is 15,511 hectares. The sample standard deviation of fixed assets' investment stock is 3323.19, and so the data numbers are large.

Table 2 statistics are made from the 2014–2017 exogenous variables, as well as the mining production stage and the land rehabilitation stage variables. The data are counted by taking the average, sample standard, maximum value, and minimum value. See Table 2 for details.

**Table 2.** Input and output variables.

| Variable | Mean | Min | Max | SD |
|---|---|---|---|---|
| Mining Employees (10,000 people) | 17.99 | 0.47 | 98.52 | 20.35 |
| Production of Non-Petroleum Mineral Resources (10,000 Tons) | 27575.75 | 931.80 | 96293.03 | 20668.17 |
| Accumulated Destruction of Land Area (Hectare) | 90093.26 | 1646.00 | 926606.00 | 168069.90 |
| Rehabilitation Investment (100 million RMB) | 40970.630 | 396.00 | 278978.00 | 46298.26 |
| Rehabilitation of Land Area (Hectare) | 1755.47 | 22.00 | 15511.00 | 2760.42 |
| Fixed Assets' Investment Stock (100 million RMB) | 4027.38 | 69.42 | 14586.76 | 3323.19 |
| Average Temperature (Centigrade) | 14.14 | 2.90 | 25.30 | 5.44 |

Note: Min - minimum; Max - maximum; SD - standard deviation.

### 4.2. Results and Analysis

Through the use of MaxDEA8.0 software, we applied the dynamic two-stage directional distance function DEA model under the exogenous variables model, to evaluate overall efficiency, mining production stage and land rehabilitation stage efficiencies, and variable efficiency.

#### 4.2.1. Overall Efficiency Analysis

The overall efficiencies of Shanxi and Fujian reached 1 in 2014 and 2016, respectively, but fell in other years, failing to maintain an efficient state. Shanxi fell from 1 to 0.652 in 2015, or the biggest drop in four years. Ningxia's annual overall efficiency was 1 in 2014 and 2017, but fell slightly in

2015–2016, from 0.797 in 2015 to 0.643 in 2016. The annual overall efficiencies of Qinghai, Hainan, and Guangxi were 1 in 2014, but fell to 0.602, 0.852, and 0.585 in 2015, respectively. In the following years, Qinghai and Guangxi recovered to different degrees, reaching 0.72 and 0.756 in 2017. The efficiency of Hainan was on the decline, falling to 0.638 in 2017. Jilin's efficiency hit 1 in 2015, but continuously declined in the following years, dropping to 0.586 in 2017. Gansu's efficiency increased significantly in 2014–2015, reaching 0.868 in 2015, but then declined in the following years, dropping to 0.749 in 2017. Jiangxi's efficiency decreased significantly from 0.668 in 2014 to 0.226. However, it rose in the years that followed, reaching 0.641 in 2017. Beijing's efficiency fluctuated sharply over four years, from 0.807 in 2014 to 0.531 in 2016, rebounding to 0.975 in 2016, but falling to 0.608 in 2017. Chongqing and Liaoning's efficiencies were low for the four years, with both at about 0.2. Overall, Chongqing was on the rise, going to 0.237 in 2017, with Liaoning having 0.296 efficiency in 2016, or the highest of these two provinces. Gansu, Xinjiang, Sichuan, Anhui, Guizhou, and other provinces tended to increase in general, but the increase was not significant. Gansu rose from 0.561 to 0.749 in 2017, which was a significant increase. Guangdong, Zhejiang, Shaanxi, Henan, and Shandong tended to decline in general and are low in the overall efficiency ranking. Among them, Shaanxi's efficiency decreased from 0.574 in 2014 to 0.304 in 2017, which is a significant decrease.

Table 3 shows the annual overall efficiency and arranges DMUs according to the overall efficiency of each province in 2014–2017.

**Table 3.** Overall efficiency for each province, from 2014 to 2017.

| Province | 2014 | 2015 | 2016 | 2017 | Province | 2014 | 2015 | 2016 | 2017 |
|---|---|---|---|---|---|---|---|---|---|
| Heilongjiang | 1 | 1 | 1 | 1 | Hubei | 0.246 | 0.28 | 0.411 | 0.315 |
| Inner Mongolia | 1 | 1 | 1 | 1 | Guangxi | 1 | 0.585 | 0.537 | 0.756 |
| Tianjin | 1 | 1 | 1 | 1 | Henan | 0.32 | 0.276 | 0.295 | 0.202 |
| Shanxi | 1 | 0.652 | 1 | 0.889 | Guizhou | 0.221 | 0.441 | 0.246 | 0.289 |
| Fujian | 1 | 0.806 | 1 | 0.659 | Shandong | 0.308 | 0.512 | 0.196 | 0.172 |
| Ningxia | 1 | 0.797 | 0.643 | 1 | Beijing | 0.807 | 0.531 | 0.975 | 0.608 |
| Gansu | 0.561 | 0.868 | 0.851 | 0.749 | Hunan | 0.408 | 0.253 | 0.176 | 0.212 |
| Hainan | 1 | 0.852 | 0.749 | 0.638 | Jiangsu | 0.349 | 0.332 | 0.172 | 0.317 |
| Xinjiang | 0.736 | 0.544 | 0.619 | 0.749 | Hebei | 0.248 | 0.176 | 0.275 | 0.244 |
| Qinghai | 1 | 0.602 | 0.661 | 0.72 | Yunnan | 0.153 | 0.218 | 0.168 | 0.353 |
| Guangdong | 0.773 | 0.715 | 0.624 | 0.667 | Shaanxi | 0.574 | 0.111 | 0.557 | 0.304 |
| Sichuan | 0.462 | 0.525 | 0.359 | 0.516 | Anhui | 0.238 | 0.12 | 0.207 | 0.321 |
| Jiangxi | 0.668 | 0.226 | 0.523 | 0.641 | Liaoning | 0.205 | 0.176 | 0.296 | 0.233 |
| Jilin | 0.658 | 1 | 0.627 | 0.586 | Chongqing | 0.177 | 0.132 | 0.211 | 0.237 |
| Zhejiang | 0.67 | 0.797 | 0.645 | 0.581 | | | | | |

From the perspective of the overall efficiency of each province, Heilongjiang, Inner Mongolia, and Tianjin had the highest overall efficiency, as their efficiency levels hit the DEA optimum. However, the overall efficiencies of Anhui, Liaoning, and Chongqing were low, and the overall efficiencies of these three provinces fluctuated around 0.2. The overall efficiencies of Qinghai, Guangdong, Sichuan, Jiangxi, and Jilin were about 0.5, and thus they need to improve. Figure 2 shows the overall efficiency of China by using the bitmap.

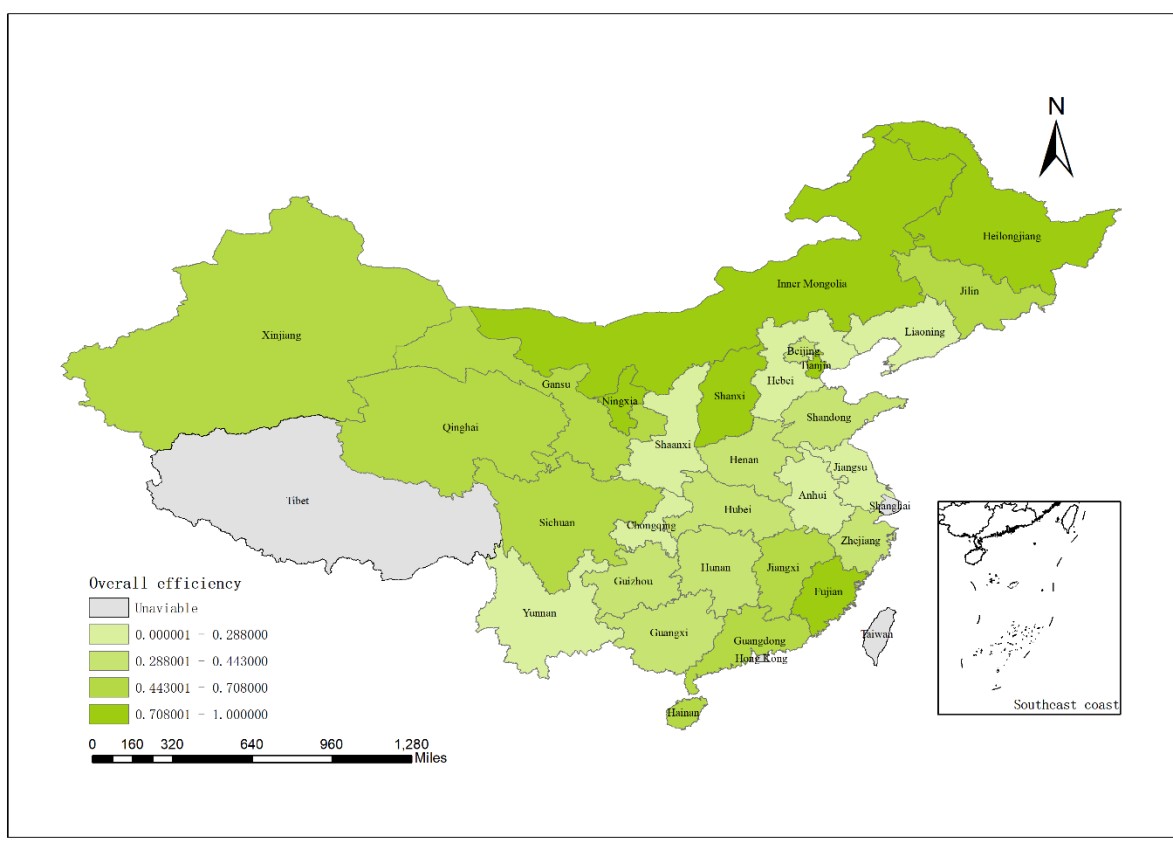

**Figure 2.** Overall efficiency for each province.

By dividing DMUs into eastern, central, and western regions, this paper shows that there are provinces with higher efficiency level and provinces with lower efficiency level in all three regions of China, and the regional internal differences are large. For example, in the eastern region, the efficiency levels of Hebei, Jiangsu, and Liaoning fluctuated around 0.2, while that of Beijing, Fujian, and Guangdong were mostly above 0.6. However, the difference between regions is not significant, and the average levels of efficiencies of the three regions were above 0.4. In 2014–2016, the efficiency of the eastern region was higher than that of the central and western regions, but in 2017, the average efficiency level of the western region was the highest.

Figure 3 analyzes the efficiency levels of Eastern, Central, and Western China in 2014–2017, by a cluster column chart and broken line chart.

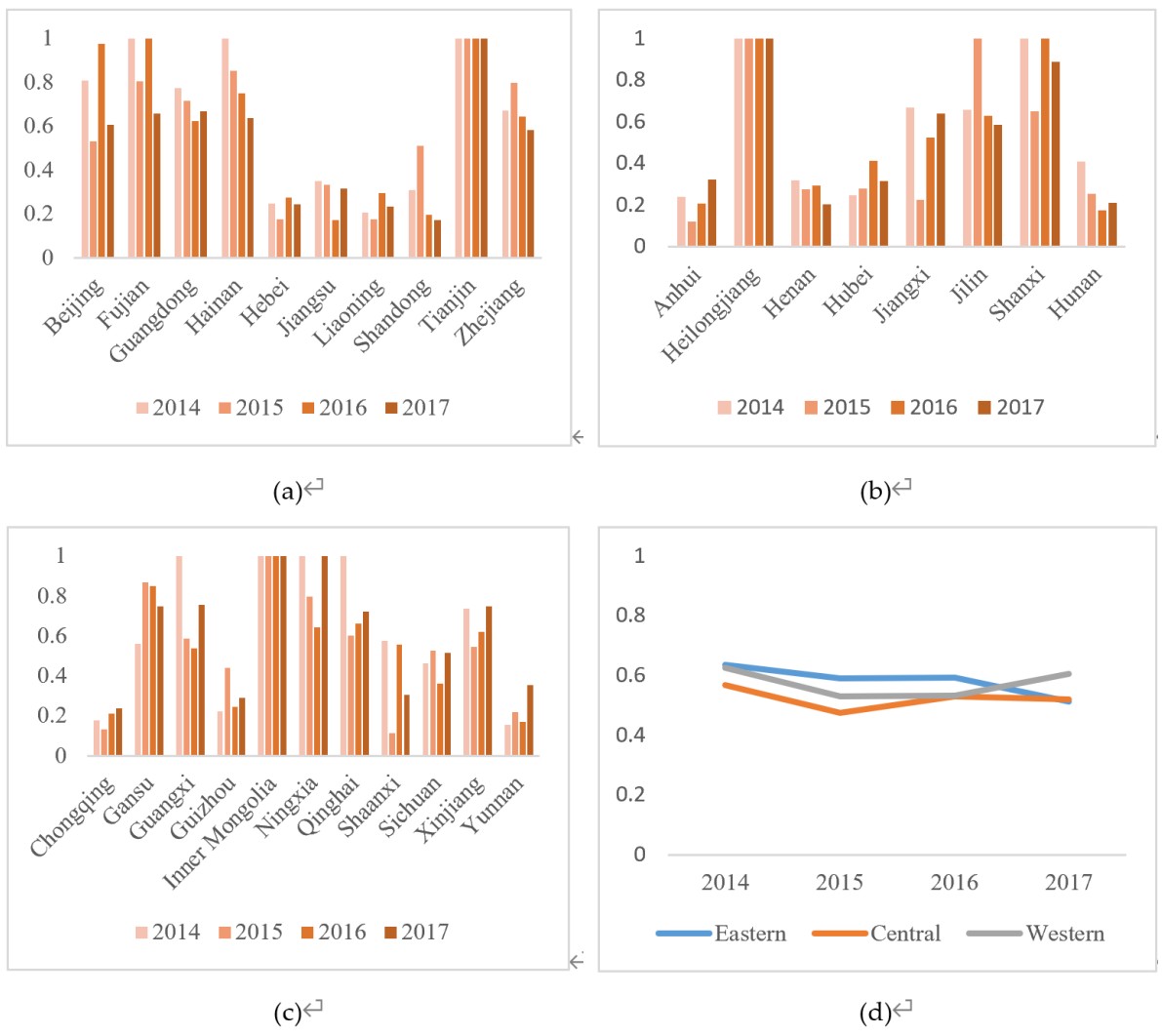

**Figure 3.** Overall efficiencies: (a) Eastern China; (b) Central China ; (c) Western China; and (d) comparison of the three regions.

#### 4.2.2. Analysis of Mining Production Stage and Land Rehabilitation Stage Efficiencies

(1)  Mining production stage efficiency

In the mining production stage, the mean efficiency of most provinces remained at a high level, among which the efficiencies of Beijing, Fujian, Guangdong, Guangxi, Hainan, Heilongjiang, Jilin, Inner Mongolia, Ningxia, Qinghai, Shanxi, Tianjin, and Zhejiang hit 1 in the four years. The efficiencies of Anhui and Hebei were below 0.2 in the four years, but continued rising to 0.147 and 0.159, respectively. The efficiencies of Henan, Shandong, Yunnan, and Hunan were low and under 0.1 in the four years. The efficiency of Henan in 2014 was 0.035, and its mean efficiency was 0.046, or the lowest value overall. Gansu's efficiency continued rising in the four years, with all values above 0.7 and maintaining a high level. It is worth noting that the efficiencies of Guizhou, Jiangsu, Hubei, Liaoning, and Chongqing were also low, fluctuating around 0.2. The efficiency of Xinjiang was more than 0.5 in the four years, but on the whole, it shows a downward trend. The mean efficiency of Shaanxi was only 0.121, and the efficiency has been declining for four years, dropping to 0.091 in 2017.

(2)  Land rehabilitation stage efficiency

In the land rehabilitation stage, the mean efficiency of most provinces remained above 0.4, showing a more balanced situation. The mean efficiencies and annual efficiencies of Heilongjiang, Inner Mongolia, and Tianjin were all 1 in the four years, showing a trend of high efficiency levels. The efficiencies of Fujian, Shaanxi, and Shanxi hit 1 in 2014 and 2016, but fluctuated dramatically in 2016 and 2017, among which the efficiency of Shaanxi sharply decreased from 1 in 2014 to 0.09 in

2015. The efficiencies in Guangxi, Hainan, and Jiangxi were 1 in 2014, but declined by different degrees in the following three years. Among them, Jiangxi's efficiency rose to 0.984 in 2017 after a decline and remained at a high level. However, the efficiency level of Guangxi declined significantly, dropping to a low level in 2016 at only 0.074. After reaching 1, the efficiency of Hainan continued declining to 0.277 in 2017. The efficiencies of Henan, Guangdong, Hunan, Shandong, and Jiangsu showed a downward trend in the four years. Hunan's efficiency dropped from 0.732 in 2014 to 0.338 in 2017. The efficiency levels of Sichuan and Xinjiang reached over 0.9 in 2017, showing high values.

Table 4 summarizes the annual efficiency and mean efficiency of each province in the mining production stage and land rehabilitation stage in 2014–2017.

**Table 4.** Mining production stage and rehabilitation stage efficiencies from 2014–2017.

| Province | M-S1 | M-S2 | 14-S1 | 14-S2 | 15-S1 | 15-S2 | 16-S1 | 16-S2 | 17-S1 | 17-S2 |
|---|---|---|---|---|---|---|---|---|---|---|
| Anhui | 0.129 | 0.315 | 0.127 | 0.350 | 0.108 | 0.132 | 0.133 | 0.282 | 0.147 | 0.495 |
| Beijing | 1.000 | 0.461 | 1.000 | 0.614 | 1.000 | 0.063 | 1.000 | 0.950 | 1.000 | 0.216 |
| Fujian | 1.000 | 0.733 | 1.000 | 1.000 | 1.000 | 0.613 | 1.000 | 1.000 | 1.000 | 0.319 |
| Gansu | 0.737 | 0.777 | 0.710 | 0.411 | 0.736 | 1.000 | 0.702 | 1.000 | 0.802 | 0.696 |
| Guangdong | 1.000 | 0.389 | 1.000 | 0.546 | 1.000 | 0.430 | 1.000 | 0.247 | 1.000 | 0.334 |
| Guangxi | 1.000 | 0.439 | 1.000 | 1.000 | 1.000 | 0.169 | 1.000 | 0.074 | 1.000 | 0.511 |
| Guizhou | 0.127 | 0.471 | 0.134 | 0.309 | 0.116 | 0.765 | 0.128 | 0.365 | 0.132 | 0.445 |
| Hainan | 1.000 | 0.620 | 1.000 | 1.000 | 1.000 | 0.705 | 1.000 | 0.497 | 1.000 | 0.277 |
| Hebei | 0.126 | 0.346 | 0.109 | 0.387 | 0.118 | 0.233 | 0.116 | 0.434 | 0.159 | 0.330 |
| Henan | 0.046 | 0.501 | 0.035 | 0.605 | 0.034 | 0.518 | 0.048 | 0.543 | 0.068 | 0.336 |
| Heilongjiang | 1.000 | 1.000 | 1.000 | 1.000 | 1.000 | 1.000 | 1.000 | 1.000 | 1.000 | 1.000 |
| Hubei | 0.147 | 0.479 | 0.148 | 0.343 | 0.138 | 0.422 | 0.138 | 0.685 | 0.165 | 0.465 |
| Hunan | 0.089 | 0.435 | 0.085 | 0.732 | 0.089 | 0.417 | 0.097 | 0.255 | 0.087 | 0.338 |
| Jilin | 1.000 | 0.435 | 1.000 | 0.316 | 1.000 | 1.000 | 1.000 | 0.253 | 1.000 | 0.172 |
| Jiangsu | 0.162 | 0.423 | 0.154 | 0.543 | 0.166 | 0.497 | 0.188 | 0.156 | 0.138 | 0.496 |
| Jiangxi | 0.249 | 0.780 | 0.336 | 1.000 | 0.148 | 0.304 | 0.214 | 0.833 | 0.297 | 0.984 |
| Liaoning | 0.198 | 0.256 | 0.205 | 0.205 | 0.200 | 0.152 | 0.210 | 0.382 | 0.179 | 0.287 |
| Inner Mongolia | 1.000 | 1.000 | 1.000 | 1.000 | 1.000 | 1.000 | 1.000 | 1.000 | 1.000 | 1.000 |
| Ningxia | 1.000 | 0.720 | 1.000 | 1.000 | 1.000 | 0.594 | 1.000 | 0.286 | 1.000 | 1.000 |
| Qinghai | 1.000 | 0.492 | 1.000 | 1.000 | 1.000 | 0.204 | 1.000 | 0.323 | 1.000 | 0.441 |
| Shandong | 0.062 | 0.533 | 0.065 | 0.552 | 0.064 | 0.960 | 0.061 | 0.331 | 0.056 | 0.288 |

| | | | | | | | | | | |
|---|---|---|---|---|---|---|---|---|---|---|
| Shanxi | 1.000 | 0.771 | 1.000 | 1.000 | 1.000 | 0.304 | 1.000 | 1.000 | 1.000 | 0.779 |
| Shaanxi | 0.121 | 0.652 | 0.148 | 1.000 | 0.132 | 0.090 | 0.115 | 1.000 | 0.091 | 0.518 |
| Sichuan | 0.134 | 0.797 | 0.152 | 0.771 | 0.135 | 0.915 | 0.118 | 0.600 | 0.129 | 0.903 |
| Tianjin | 1.000 | 1.000 | 1.000 | 1.000 | 1.000 | 1.000 | 1.000 | 1.000 | 1.000 | 1.000 |
| Xinjiang | 0.565 | 0.759 | 0.626 | 0.846 | 0.530 | 0.558 | 0.563 | 0.676 | 0.540 | 0.957 |
| Yunnan | 0.078 | 0.368 | 0.078 | 0.228 | 0.079 | 0.358 | 0.080 | 0.256 | 0.076 | 0.631 |
| Zhejiang | 1.000 | 0.346 | 1.000 | 0.340 | 1.000 | 0.594 | 1.000 | 0.289 | 1.000 | 0.162 |
| Chongqing | 0.158 | 0.221 | 0.186 | 0.169 | 0.124 | 0.141 | 0.154 | 0.269 | 0.168 | 0.305 |

Note: S1 - mining production stage; S2 - land rehabilitation stage; M -mean value.

The efficiencies of the two stages in all provinces of China were, on the whole, above 0.5. The average efficiency of the land rehabilitation stage was slightly higher than the mining production stage, and the efficiencies of these two stages were relatively balanced. Based on the average efficiency of each province, we divide the areas into four parts: high-high, low-low, high-low, and low-high. Among them, there are many high-high and low-low areas, showing serious polarization in the mining production stage and land rehabilitation stage input–output efficiencies. Comparing the high-low and low-high areas, there are more provinces in high-low areas, including Beijing, Guangdong, Guangxi, Jilin, Qinghai, and Zhejiang. The mining production stage efficiency is higher, while the land rehabilitation stage efficiency is below average. See Figure 4 for the distribution of specific provinces.

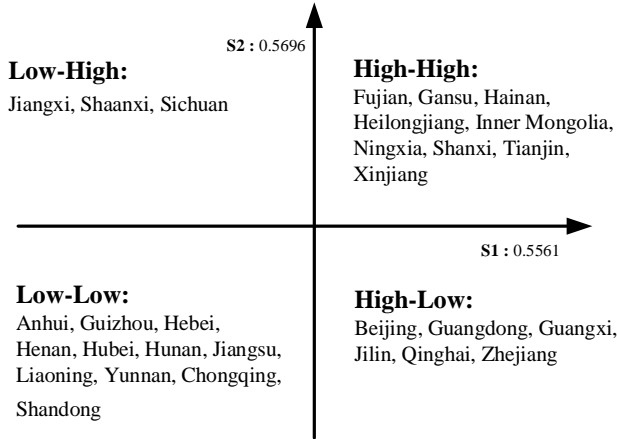

**Figure 4.** Province distribution by mining production and land rehabilitation stages.

There were great differences in the efficiencies of different stages in the provinces of China. There were also great fluctuations among the provinces in each year, which closely relate to the mineral resources of each region and the measures taken. In the four years, the efficiency of the mining production stage was higher than that of the land rehabilitation stage.

In the mining production stage, the level of DEA hit 1, but in the land rehabilitation stage, the efficiency was less than 0.5, including Beijing, Hainan, Jilin, and Qinghai. Figure 5 applies a radar map to describe the mining production stage and land rehabilitation stage efficiencies of each province.

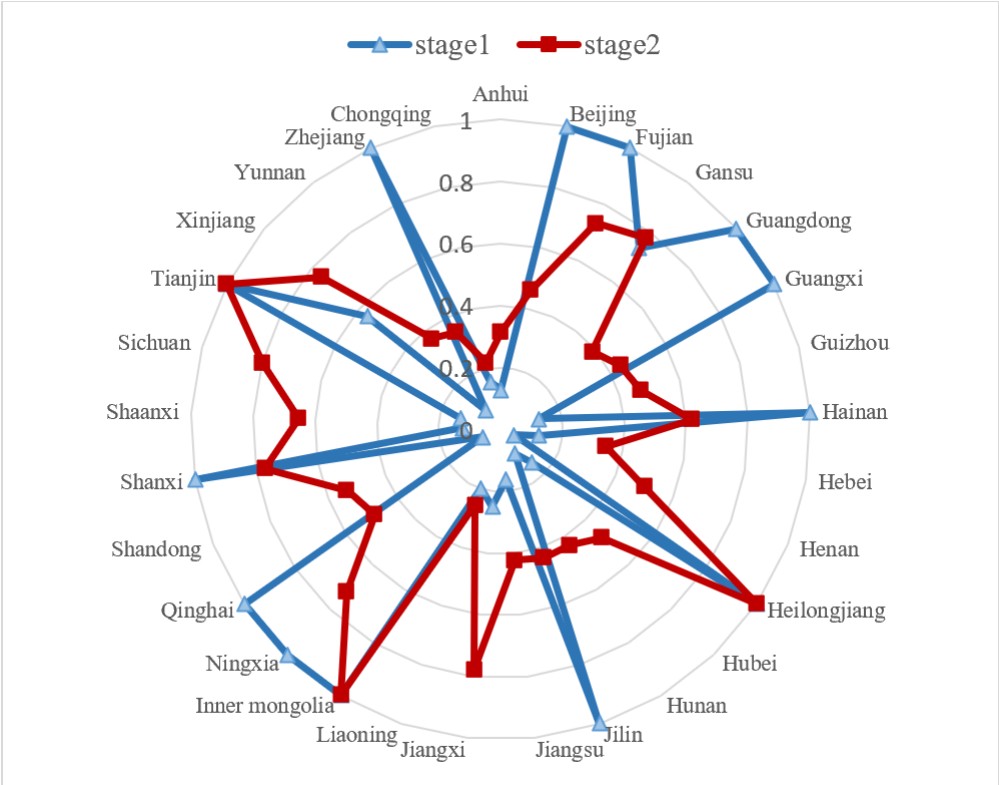

**Figure 5.** Regional overall stage efficiency.

### 4.2.3. Efficiency Analysis of Input and Output Variables

In the input of the mining production stage, the efficiencies of mining employees in most provinces were high, among which the efficiencies of Beijing, Fujian, Guangdong, Guangxi, Hainan, Heilongjiang, Jilin, Inner Mongolia, Ningxia, Qinghai, Shanxi, Tianjin, and Zhejiang were all 1 in the four years. In the mining production stage, the production efficiencies of non-petroleum mineral resources in most provinces were high. The efficiencies of Beijing, Fujian, Guangdong, Guangxi, Hainan, Jilin, Inner Mongolia, Ningxia, Shanxi, Tianjin, and Zhejiang all remained at 1. However, the efficiencies of Hubei, Hunan, and Jiangsu were relatively low, among which the efficiency of Jiangsu continued declining in the four years, arriving at 0.292 in 2017. The efficiency in Jiangxi fluctuated dramatically in the four years, with an efficiency of 1 in 2014 and 2016, but it dropped to 0.622 in 2017. Liaoning's efficiency kept above 0.9 in 2014–2016, but dropped to 0.579 in 2017, which is a significant decline. In general, the efficiencies of Sichuan and Yunnan showed a downward trend, respectively falling to 0.61 and 0.482 in 2017. Except for 2014, when the efficiency of Chongqing remained at 1, the efficiencies of the other years fluctuated around 0.3.

In the mining production stage, the accumulated destruction of land area is the undesirable output. Since the accumulated destruction of land area is a special connecting variable of the two stages, the efficiencies of most provinces remained at 1, showing a high level. However, the efficiency of this variable was relatively low in Chongqing, at under 0.2 in 2015-2017. The efficiency of Jiangxi was 1 in 2014, but then decreased to 0.418 in 2015, for a significant decline.

Table 5 lists the output variables in the mining production stage, the production amount of non-petroleum mineral resources, and the accumulated destruction of land area.

**Table 5.** Input and output efficiencies in the mining production stage.

| Province | ME-14 | ME-15 | ME-16 | ME-17 | MP-14 | AD-14 | MP-15 | AD-15 | MP-16 | AD-16 | MP-17 | AD-17 |
|---|---|---|---|---|---|---|---|---|---|---|---|---|
| Anhui | 0.127 | 0.108 | 0.133 | 0.147 | 0.999 | 1.000 | 1.000 | 0.586 | 1.000 | 1.000 | 1.000 | 1.000 |
| Beijing | 1.000 | 1.000 | 1.000 | 1.000 | 1.000 | 1.000 | 1.000 | 1.000 | 1.000 | 1.000 | 1.000 | 1.000 |
| Fujian | 1.000 | 1.000 | 1.000 | 1.000 | 1.000 | 1.000 | 1.000 | 1.000 | 1.000 | 1.000 | 1.000 | 1.000 |
| Gansu | 0.710 | 0.736 | 0.702 | 0.802 | 1.000 | 1.000 | 1.000 | 1.000 | 1.000 | 1.000 | 1.000 | 1.000 |
| Guangdong | 1.000 | 1.000 | 1.000 | 1.000 | 1.000 | 1.000 | 1.000 | 1.000 | 1.000 | 1.000 | 1.000 | 1.000 |
| Guangxi | 1.000 | 1.000 | 1.000 | 1.000 | 1.000 | 1.000 | 1.000 | 1.000 | 1.000 | 1.000 | 1.000 | 1.000 |
| Guizhou | 0.134 | 0.116 | 0.128 | 0.132 | 0.862 | 1.000 | 0.771 | 1.000 | 0.693 | 1.000 | 0.751 | 1.000 |
| Hainan | 1.000 | 1.000 | 1.000 | 1.000 | 1.000 | 1.000 | 1.000 | 1.000 | 1.000 | 1.000 | 1.000 | 1.000 |
| Hebei | 0.109 | 0.119 | 0.116 | 0.159 | 0.960 | 1.000 | 0.999 | 1.000 | 0.999 | 1.000 | 0.794 | 1.000 |
| Henan | 0.035 | 0.034 | 0.048 | 0.068 | 0.926 | 1.000 | 0.710 | 0.655 | 0.998 | 1.000 | 0.999 | 0.972 |
| Heilongjiang | 1.000 | 1.000 | 1.000 | 1.000 | 1.000 | 1.000 | 1.000 | 1.000 | 1.000 | 1.000 | 1.000 | 1.000 |
| Hubei | 0.149 | 0.138 | 0.138 | 0.165 | 0.442 | 1.000 | 0.509 | 1.000 | 0.388 | 1.000 | 0.420 | 1.000 |
| Hunan | 0.085 | 0.089 | 0.097 | 0.087 | 0.551 | 1.000 | 0.653 | 1.000 | 0.522 | 1.000 | 0.443 | 1.000 |
| Jilin | 1.000 | 1.000 | 1.000 | 1.000 | 1.000 | 1.000 | 1.000 | 1.000 | 1.000 | 1.000 | 1.000 | 1.000 |
| Jiangsu | 0.154 | 0.166 | 0.188 | 0.138 | 0.524 | 1.000 | 0.510 | 1.000 | 0.357 | 1.000 | 0.292 | 1.000 |
| Jiangxi | 0.336 | 0.148 | 0.214 | 0.297 | 1.000 | 1.000 | 0.752 | 0.418 | 1.000 | 0.754 | 0.622 | 0.617 |
| Liaoning | 0.205 | 0.200 | 0.210 | 0.179 | 0.923 | 1.000 | 0.999 | 1.000 | 0.999 | 1.000 | 0.579 | 0.713 |
| Inner Mongolia | 1.000 | 1.000 | 1.000 | 1.000 | 1.000 | 1.000 | 1.000 | 1.000 | 1.000 | 1.000 | 1.000 | 1.000 |
| Ningxia | 1.000 | 1.000 | 1.000 | 1.000 | 1.000 | 1.000 | 1.000 | 1.000 | 1.000 | 1.000 | 1.000 | 1.000 |
| Qinghai | 1.000 | 1.000 | 1.000 | 1.000 | 1.000 | 1.000 | 1.000 | 1.000 | 1.000 | 1.000 | 1.000 | 1.000 |
| Shandong | 0.065 | 0.065 | 0.061 | 0.056 | 0.886 | 1.000 | 0.999 | 1.000 | 0.999 | 1.000 | 0.679 | 1.000 |
| Shanxi | 1.000 | 1.000 | 1.000 | 1.000 | 1.000 | 1.000 | 1.000 | 1.000 | 1.000 | 1.000 | 1.000 | 1.000 |
| Shaanxi | 0.148 | 0.132 | 0.115 | 0.091 | 0.928 | 1.000 | 0.999 | 1.000 | 0.999 | 1.000 | 0.999 | 1.000 |
| Sichuan | 0.152 | 0.135 | 0.118 | 0.129 | 0.908 | 1.000 | 0.745 | 1.000 | 0.646 | 1.000 | 0.610 | 1.000 |
| Tianjin | 1.000 | 1.000 | 1.000 | 1.000 | 1.000 | 1.000 | 1.000 | 1.000 | 1.000 | 1.000 | 1.000 | 1.000 |
| Xinjiang | 0.626 | 0.530 | 0.563 | 0.540 | 1.000 | 1.000 | 1.000 | 1.000 | 1.000 | 1.000 | 1.000 | 1.000 |
| Yunnan | 0.078 | 0.079 | 0.080 | 0.076 | 0.752 | 1.000 | 0.815 | 1.000 | 0.780 | 1.000 | 0.482 | 0.629 |
| Zhejiang | 1.000 | 1.000 | 1.000 | 1.000 | 1.000 | 1.000 | 1.000 | 1.000 | 1.000 | 1.000 | 1.000 | 1.000 |
| Chongqing | 0.186 | 0.124 | 0.154 | 0.168 | 1.000 | 0.595 | 0.358 | 0.112 | 0.270 | 0.130 | 0.301 | 0.164 |

Note: MP - production of non-petroleum mineral resources; AD - accumulated destruction of land area; ME - mining employees.

In the input variables of the land rehabilitation stage, the efficiencies of Heilongjiang, Inner Mongolia, and Tianjin were all 1 in the four years. In 2014 and 2016, the efficiencies of Fujian, Shanxi, and Shaanxi reached 1, but in 2015 and 2017 the efficiency of Fujian dropped to 0.189 in 2015, or the most serious drop among the three provinces. The efficiencies of Hainan and Gansu were higher at above 0.9 in the four years, while that of Gansu was 1 in 2015 and 2016. Guangxi's efficiency was 1 in 2014, but continued declining in the following two years, rebounding to 0.949 in 2017. The efficiency of Guizhou fell significantly in 2014–2015, but recovered to above 0.9 in 2016–2017. Zhejiang's efficiency continued declining in 2014–2016, but rebounded to 0.956 in 2017. The efficiencies of Liaoning, Chongqing, Henan, and Guangdong were relatively high at about 0.9 in 2014–2017. Chongqing's efficiency in 2014 was 0.847, or the lowest among the four provinces.

In the output variables of the land rehabilitation stage, the efficiencies of most provinces were below 0.5. Among them, the efficiencies of Fujian, Shaanxi, and Shanxi were similar to the efficiency of rehabilitation investment. In 2014 and 2016, their efficiencies hit 1. The efficiencies for the abovementioned provinces in 2015 and 2017 decreased, but the efficiency of the rehabilitation of the land area variable decreased more dramatically. In 2015, the efficiency of Shaanxi decreased to 0.103. The efficiencies of Anhui and Chongqing were poor compared with the efficiency of rehabilitation investment, and their efficiencies in 2015 were 0.144 and 0.15, respectively. Thus, Anhui should pay attention to improving the efficiency of the output stage. The efficiencies in Guangxi and Hainan hit 1 in 2014, but declined in the following years, with the efficiency in Guangxi going to 0.1 in 2016. Beijing's efficiency fluctuated in four years, with a trough of 2015 when the efficiency was only 0.072 and a peak of 2016 when the efficiency was 0.957.

Table 6 summarizes the input and output variables of the land rehabilitation stage, including rehabilitation investment and rehabilitation of land area.

**Table 6.** Input and output efficiencies in the land rehabilitation stage.

| Province | RI-14 | RA-14 | RI-15 | RA-15 | RI-16 | RA-16 | RI-17 | RA-17 |
|---|---|---|---|---|---|---|---|---|
| Anhui | 0.955 | 0.366 | 0.915 | 0.144 | 0.918 | 0.307 | 0.946 | 0.523 |
| Beijing | 0.962 | 0.638 | 0.873 | 0.072 | 0.993 | 0.957 | 0.923 | 0.234 |
| Fujian | 1.000 | 1.000 | 0.189 | 0.616 | 1.000 | 1.000 | 0.965 | 0.330 |
| Gansu | 0.952 | 0.432 | 1.000 | 1.000 | 1.000 | 1.000 | 0.970 | 0.718 |
| Guangdong | 0.965 | 0.565 | 0.940 | 0.457 | 0.910 | 0.272 | 0.951 | 0.352 |
| Guangxi | 1.000 | 1.000 | 0.886 | 0.191 | 0.742 | 0.100 | 0.949 | 0.539 |
| Guizhou | 0.478 | 0.314 | 0.187 | 0.768 | 0.955 | 0.383 | 0.969 | 0.460 |
| Hainan | 1.000 | 1.000 | 0.970 | 0.727 | 0.907 | 0.548 | 0.931 | 0.297 |
| Hebei | 0.723 | 0.396 | 0.459 | 0.241 | 0.840 | 0.449 | 0.929 | 0.355 |
| Henan | 0.969 | 0.625 | 0.948 | 0.547 | 0.936 | 0.579 | 0.927 | 0.363 |
| Heilongjiang | 1.000 | 1.000 | 1.000 | 1.000 | 1.000 | 1.000 | 1.000 | 1.000 |
| Hubei | 0.720 | 0.351 | 0.254 | 0.427 | 0.973 | 0.705 | 0.965 | 0.482 |
| Hunan | 0.572 | 0.737 | 0.548 | 0.429 | 0.399 | 0.260 | 0.955 | 0.354 |

| | | | | | | | | |
|---|---|---|---|---|---|---|---|---|
| Jilin | 0.936 | 0.338 | 1.000 | 1.000 | 0.850 | 0.298 | 0.913 | 0.188 |
| Jiangsu | 0.705 | 0.552 | 0.264 | 0.503 | 0.926 | 0.169 | 0.968 | 0.512 |
| Jiangxi | 1.000 | 1.000 | 0.917 | 0.332 | 0.968 | 0.860 | 0.998 | 0.985 |
| Liaoning | 0.948 | 0.216 | 0.923 | 0.164 | 0.837 | 0.456 | 0.928 | 0.309 |
| Inner Mongolia | 1.000 | 1.000 | 1.000 | 1.000 | 1.000 | 1.000 | 1.000 | 1.000 |
| Ningxia | 1.000 | 1.000 | 0.927 | 0.641 | 0.817 | 0.350 | 1.000 | 1.000 |
| Qinghai | 1.000 | 1.000 | 0.932 | 0.219 | 0.867 | 0.372 | 0.942 | 0.468 |
| Shandong | 0.614 | 0.560 | 0.489 | 0.961 | 0.951 | 0.348 | 0.944 | 0.306 |
| Shanxi | 1.000 | 1.000 | 0.933 | 0.326 | 1.000 | 1.000 | 0.979 | 0.795 |
| Shaanxi | 1.000 | 1.000 | 0.873 | 0.103 | 1.000 | 1.000 | 0.964 | 0.537 |
| Sichuan | 0.987 | 0.781 | 0.336 | 0.918 | 0.971 | 0.618 | 0.990 | 0.912 |
| Tianjin | 1.000 | 1.000 | 1.000 | 1.000 | 1.000 | 1.000 | 1.000 | 1.000 |
| Xinjiang | 0.986 | 0.858 | 0.954 | 0.585 | 0.939 | 0.719 | 0.995 | 0.962 |
| Yunnan | 0.664 | 0.234 | 0.700 | 0.372 | 0.928 | 0.276 | 0.960 | 0.657 |
| Zhejiang | 0.613 | 0.347 | 0.307 | 0.600 | 0.549 | 0.298 | 0.956 | 0.170 |
| Chongqing | 0.847 | 0.199 | 0.938 | 0.150 | 0.855 | 0.315 | 0.931 | 0.328 |

Note: RI - rehabilitation investment; RA - rehabilitation of land.

*4.3. Policy Analysis*

4.3.1. Efficiency of Special Provinces

With the transfer of Beijing's non-capital functions, its mining industry has gradually shut down. In this process, the accumulated destruction of land area in Beijing remained basically unchanged. In 2015, the efficiency in Beijing fell to a low level, and in 2016 it rose to a peak. This is closely related to the lag in the process of land destruction. Generally speaking, in the land rehabilitation stage, it is still necessary to improve the historical land destruction in Beijing by strengthening land rehabilitation efficiency. The efficiencies of Shanxi and Shaanxi decreased significantly in 2015 and 2017, which closely relate to the low efficiencies of rehabilitation investment and rehabilitation of land area variables in the land rehabilitation stage. As they are provinces with large mining resources, although the mining production stages of Shanxi and Shaanxi in 2014–2017 have high levels of output variable efficiency, they must strengthen remediation in the land rehabilitation stage and thus help promote overall efficiency.

The efficiency of Jiangxi tended to rise in 2014–2017, but in the mining production stage, its efficiency was poor, thus dragging down its overall efficiency. The regulations of Jiangxi Province on the management of the mining of mineral resources issued in 2015 clearly planned for the protection and restoration of the geological environments of mines, supervision and management, and legal responsibility, and so the efficiency of Jiangxi in the land rehabilitation stage remained at a high level. However, in the mining stage, Jiangxi should pay attention to the improvement of input and output variables' efficiencies and promote the improvement of the overall level of efficiency. Jilin's efficiency

declined sharply in 2017, which is closely related to the sharp decline of its land rehabilitation efficiency in 2015. It can be seen during the process of mining production and land rehabilitation that, although mining production stage efficiency is high, some variables in the land rehabilitation stage need to improve their efficiency level to enhance overall efficiency. Due to the low quality of mines in Hebei, the efficiency in the mining production stage is low. To undertake the process of Beijing's non-capital function, the supply of primary mineral products is in great demand, which promotes the quantity of mining in Hebei, but ignores the efficiency problem in the mining process. Although land rehabilitation stage efficiency is higher than mining production stage efficiency, the former is still at a low level.

### 4.3.2. Improvement of Each Province

In order to distinguish between the variables that need to be improved in each province, we set 0~0.33 as grade I, 0.33~0.66 as grade II, and 0.66~1 as grade III. Grade I means the indicator needs more focused improvement, grade II means the indicator needs improvement, and grade III means the need for slight improvement or no need for improvement. The provinces that do not need to be improved include Fujian, Gansu, Heilongjiang, Inner Mongolia, Ningxia, Shanxi, and Tianjin; Guizhou, Hebei, Henan, Hubei, Hunan, Liaoning, and other provinces need to focus on improvement and strengthen their overall governance according to the local situation; and Chongqing needs to focus on mining production and land rehabilitation as a whole, among which accumulated destruction of land area and rehabilitation area need focused improvement, and one variable needs improvement, which is the production of mineral resources. Table 7 shows the room for improvement of provinces in more detail.

**Table 7.** Improvement for each province.

| Province | MP | AD | RI | RA | Province | MP | AD | RI | RA |
|---|---|---|---|---|---|---|---|---|---|
| Anhui | III | III | III | II | Jiangxi | III | III | III | III |
| Beijing | III | III | III | II | Liaoning | III | III | III | I |
| Fujian | III | III | III | III | Inner Mongolia | III | III | III | III |
| Gansu | III | III | III | III | Ningxia | III | III | III | III |
| Guangdong | III | III | III | II | Qinghai | III | III | III | II |
| Guangxi | III | III | III | II | Shandong | III | III | III | II |
| Guizhou | III | III | II | II | Shanxi | III | III | III | III |
| Hainan | III | III | III | II | Shaanxi | III | III | III | III |
| Hebei | III | III | III | II | Sichuan | III | III | III | III |
| Henan | III | III | III | II | Tianjin | III | III | III | III |
| Heilongjiang | III | III | III | III | Xinjiang | III | III | III | III |
| Hubei | II | III | III | II | Yunnan | III | III | III | II |
| Hunan | II | III | II | II | Zhejiang | III | III | II | II |
| Jilin | III | III | III | II | Chongqing | II | I | III | I |

| Jiangsu | II | III | III | II |
| --- | --- | --- | --- | --- |

Note: I: 0~0.33; II: 0.33~0.66; III: 0.66~1

## 5. Conclusion and Policy Recommendations

### 5.1. Conclusion

This research uses the dynamic two-stage directional distance function DEA model, with environmental exogenous variables, to measure the mining production/land rehabilitation efficiencies of 29 provinces in China and arrived at the following conclusions.

The overall efficiencies of most provinces in China are in general below 0.5. Among them, the efficiencies of Tianjin, Inner Mongolia, and Heilongjiang reached the DEA optimal level, which closely relate to the distribution of local natural resources and policy guidance. The efficiencies of Anhui, Chongqing, Jiangsu, and Liaoning were relatively low. There was a big fluctuation in the efficiency of the land rehabilitation stage in the four years, but the average efficiency was slightly higher than that in the mining production stage. Thus, the fluctuation of land restoration efficiency should be reduced to make it grow steadily better. For the process of demand-and-supply transformation of the mining industry, China still faces greater risks and problems that must be improved.

By comparing the efficiencies of the mining production stage and the land rehabilitation stage, we find that the efficiency of the former is relatively low. Thus, mining efficiency should urgently be improved in order to promote the overall efficiency of a region.

The efficiency distribution of China's provinces is not uniform, which relates to regional policy and resource distribution. For overall efficiency levels, Jiangsu, Anhui, Shaanxi, and Chongqing are lower, while Inner Mongolia, Heilongjiang, and Tianjin are higher.

This paper employs a theoretical analysis of 29 provinces' mining efficiency in China. The aim is to effectively guide each province in mining and rehabilitation and to deal with the lack of specialization in China for this sector, presently and in future. While this research paper's sample selection is scientific and objective, it fails to fully simulate the various emergencies of mining and land restoration.

### 5.2. Policy Recommendations

Due to the great difference between the resource distribution and efficiencies of provinces in China, the projects for improving the efficiency level of each province are different. Thus, corresponding policies need to strengthen these efficiencies. Based on the development efficiency for the period 2014–2017, the following suggestions are made.

5.2.1. Integrate Mining Resources

Through the market guidance mechanism, developed provinces should be able to control their mining output and promote the sustainable development of mining areas. Speeding up the business of mining area integration can help to gradually replace the situation of decentralized mining of small- and medium-sized mines in China. Integrating regional resources can spur a region to achieve a good state of economies of scale and scope. By integrating resources, mining areas with similar resource reserves and quality can share technology, machinery, and equipment and thus reduce repeated investment in fixed assets. This is also conducive to the spillover effect in the land rehabilitation stage, so that mining areas with a similar environment and damage conditions can reduce rehabilitation investment and obtain better rehabilitation of land area. China should continue to promote the improvement of efficiency in the land rehabilitation stage. Provinces with low efficiency levels, like Shaanxi, can learn from provinces with a similar climate environment and mining-area types, introduce advanced mining and land rehabilitation technology locally, and implement technology improvement so as to increase their level of efficiency. The government should strengthen with the surrounding water, circuit, and other infrastructure. On the premise of not

affecting the life of residents around the area, the government can give priority to providing cheap hydropower resources and mine-road subsidies for mining areas. Integration of surrounding infrastructure can achieve the purpose of resource integration around a mining area.

### 5.2.2. Eliminate Backward Capacity and Technology

Mining firms can phase out backward production capacity and replace old existing fixed assets by introducing high-tech mining machines and accessories. At the same time, they can set up corresponding research bases according to the unique situation locally, increase R&D investment, strengthen their scientific and technological levels, change from rough mining to scientific mining, and conform to the trend of high-quality development in China. In addition, a special group for land rehabilitation should set up a scientific and technological research base. By analyzing the local mining environment and soil conditions, damaged or occupied land can be restored into a historical mining park or reclaimed as cultivated land, in order to give full play to the residual value of damaged soil. Such actions can strengthen the efficiency of the land rehabilitation stage.

In view of the amount of seriously damaged land, biochemical measures should be introduced in each area, to gradually repair damaged land and improve the environment of mining areas by building a stable ecosystem. For the mountainous provinces of Guizhou, Sichuan, and Chongqing, environmental restoration should be planned according to the geological conditions of mines in each area. For the land restoration of mining areas in mountainous regions, crops with economic value and in line with China's market, such as artemisia argyi or Conyza canadensis (L.) Cronq., can be selected as vegetation restoration options in the southwest part of the country. Depending on varying slopes, different vegetation can be adopted to firm up the slopes and prevent the occurrence of landslides, so as to ensure smooth mining and land restoration. For winter freezing conditions in areas like Xinjiang, Heilongjiang, Jilin, Liaoning, and Inner Mongolia, it is necessary to solve the problem of frozen soil. By introducing new winter mining technology, the impact of bad weather on mining should be able to slow down. For the current operation of mining machines, it is possible to retrieve the braking and driving devices of crushers or excavators and clean their internal widgets after each use, especially those that are easily damaged by freezing. Mining firms must pay attention to the temperature while cleaning and check whether the widgets are leaking.

### 5.2.3. Strengthen Government Administrative Measures

For serious damage caused by mining, local governments should intervene in administrative management and guide the coordinated development of mining production and land rehabilitation. National and local people's congressional bodies should enhance the laws and regulations on the examination and approval of mining licenses, the handling of safety production licenses, and the division of responsibility for mining accidents and further guide the coordinated development of regional production and environment. As to the mining sector, each province can give appropriate tax exemptions to mining enterprises that run under high efficiency and strong land rehabilitation efficiency according to the situation of their own conditions. Through the leading role of excellent mining enterprises, the efficiency of land rehabilitation can be improved. For some old mining areas that are short of funds, each province should set up special cash reserves that can support old mining areas to update their own equipment and strengthen their efficiency level. In order to improve the use of special funds for the environmental restoration of mining areas in each province, local governments should closely track the whereabouts of the funds and present restoration results in real time, thus curbing any misappropriation of money or financial corruption. Implementing these measures should help China clean up the land damage caused by mining in local mining areas and promote the improvement of overall efficiency in this important industrial sector.

**Author Contributions:** All authors have read and agree to the published version of the manuscript. Conceptualization, Z.S.; data curation, Y.W.; formal analysis, Z.S. and Y.-H.C.; funding acquisition, Z.S.; methodology, Y.-H.C.; project administration, Z.S. and F.W.; supervision, C.S. F.W., and Y.W.; visualization, F.W. and C.S.; writing—original draft, C.S., F.W., and Y.W.; writing—review and editing, Z.S. and Y.-H.C. All authors have read and agreed to the published version of the manuscript.

**Funding:** This research was funded by the Ministry of Education Humanistic and Social Science Research Youth Funds, grant number 19YJC790112; Fundamental Research Funds for the Central Universities, grant number 2020QG1206; the National Natural Science Foundation of China, grant number 41701613.

**Conflicts of Interest:** The authors declare no conflicts of interest.

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
