# Peer review of "Dynamic Linkages among Mining Production and Land Rehabilitation Efficiency in China"

_land, doi:10.3390/land9030076_

Round 1
Reviewer 1 Report
The proposed research «Dynamic linkages among mining production and land rehabilitation efficiency in China » falls within the scope of Land. According to the reviewer’s opinion, minor revisions are required in order to accept this research study for publication in Land. Please, comply with the following suggestions and comments:
Comment 1: The paper is in general well accompanied of clear explanations. I think that some additional figures would help to the better analysis of the subject.
Comment 2: Finally, when you submit the corrected version, please do check thoroughly, in order to avoid grammar, syntax or structure/presentation flaws - please seek for professional English proofreading services or ask a native English-speaking colleague of yours in order to refine and improve the English in your paper.
Author Response
Responses to Review 1
Thank you very much for your helpful feedback and insightful comments. We have taken all of the suggestions and comments into consideration during this revision. We truly appreciate the opportunity to revise our paper, and believe that our manuscript has significantly improved in response to the ideas and recommendations of the review team.
Detailed responses:
According to the reviewer’s opinion, minor revisions are required in order to accept this research study for publication in Land. Please, comply with the following suggestions and comments:
Our Response: We greatly appreciate your detailed and specific comments, which help us improve both the rigor and the clarity of the paper. Below we address your concerns in the order they appear in your review report, and discuss how we incorporate them into the revision.
- The paper is in general well accompanied of clear explanations. I think that some additional figures would help to the better analysis of the subject.
Our Response: Thank you for pointing this out. We've added a bitmap of China's overall efficiency, and we also use the Clustered column chart and line chart to analyze the efficiency level of Eastern, central and Western China. This will help us to discuss the overall efficiency of China and the differences between regions. Details are as follows.
From the perspective of the overall efficiency of each province, Heilongjiang, Inner Mongolia, and Tianjin had the highest overall efficiency, as their efficiency levels hit the DEA optimum. However, the overall efficiencies of Anhui, Liaoning, and Chongqing were low, and the overall efficiencies of these three provinces fluctuated around 0.2. The overall efficiencies of Qinghai, Guangdong, Sichuan, Jiangxi, and Jilin were about 0.5, and thus they need to improve. Figure 2 shows the overall efficiency of China by using the bitmap.
Figure 2. Overall efficiency for each province
By dividing DMUs into eastern, central, and western regions, this paper shows that there are provinces with higher efficiency level and provinces with lower efficiency level in all three regions of China, and the regional internal differences are large. For example, in the eastern region the efficiency levels of Hebei, Jiangsu, and Liaoning fluctuated around 0.2, while that of Beijing, Fujian, and Guangdong were mostly above 0.6. However, the difference between regions is not significant, and the average levels of efficiencies of the three regions were above 0.4. In 2014-16, the efficiency of the eastern region was higher than that of the central and western regions, but in 2017 the average efficiency level of the western region was the highest.
Figure 3 analyzes the efficiency levels of eastern, central, and western China in 2014-17 by a cluster column chart and broken line chart.
|
(a) |
(b) |
|
(c) |
(d) |
Figure 3. Overall efficiencies. (a) Eastern; (b) Central; (c) Western; (d) Comparison of the three regions.
2.Finally, when you submit the corrected version, please do check thoroughly, in order to avoid grammar, syntax or structure/presentation flaws - please seek for professional English proofreading services or ask a native English-speaking colleague of yours in order to refine and improve the English in your paper.
Our Response: Thank you for pointing this out. We have sought the professional English proofreading services to to ensure the correctness of grammatical expressions.

Reviewer 2 Report
The authors assess efficiency of mining production-land rehabilitation in some provinces of China.
In Section 4.1.2. Data Descriprion, we recommend the authors to revise the paragrahps with the presentation of the variables. Morover, it is necessary to present the units of measure for all the variables.
In Section 4.2. Results, it is necessary to specify the software used for data treatment.
We reccomend the authors to revise the titles of the chapters 4.2 and 4.3.
Author Response
Responses to Review 2
Thank you very much for your helpful feedback and insightful comments. We have taken all of the suggestions and comments into consideration during this revision. We truly appreciate the opportunity to revise our paper, and believe that our manuscript has significantly improved in response to the ideas and recommendations of the review team.
Detailed responses:
- In Section 4.1.2. Data Description, we recommend the authors to revise the paragraphs with the presentation of the variables. Moreover, it is necessary to present the units of measure for all the variables.
Our Response: Thank you for pointing this out. we have reorganized the presentation of 4.1.2 and add units to make it more reasonable. Details are as follows.
4.1.2. Data description
The average temperature in various regions of China is influenced by latitude factors. The average temperature is 25.3℃, while the minimum temperature is only 2.9℃, and so there is differentiated regional distribution. However, due to the particularity of average temperature factors, the standard deviation is 5.441, and the temperature horizontal differences between regions are low. The maximum number of mining employees is 0.985 million. In this variable, the standard deviation is low at only is 20.346. The average production of non-petroleum mineral resources is 275.7 million tons, the maximum value is 962.9 million tons, and the minimum value is 9.3 million tons, greatly correlating to the distribution of natural resources in various regions. The maximum value of accumulated destruction land area is 926,606 hectares, and the minimum value is 1,646 hectares. The average amount of rehabilitation investment is 4,097,063 million RMB, and the sample standard deviation is 46,298, closely correlating to the accumulated destruction of land area in various regions. The average rehabilitation of land area is 1,755.4 hectares, while the minimum value is only 22 hectares, and the maximum value is 15,511 hectares. The sample standard deviation of fixed assets’ investment stock is 3,323.1, and so the data numbers are large.
- In Section 4.2. Results, it is necessary to specify the software used for data treatment.
Our Response: Thank you for pointing this out. We've specified the software used for data treatment. This is on page 12. Details are as follows.
4.2. Results and analysis
Through the use of MaxDEA8.0 software, we apply the dynamic two-stage directional distance function DEA model under the exogenous variables model to evaluate overall efficiency, mining production stage and land rehabilitation stage efficiencies, and variable efficiency.
- We recommend the authors to revise the titles of the chapters 4.2 and 4.3.
Our Response: Thank you for pointing this out. In order to make it more suitable for chapter content, we have changed the title of 4.3 from “discussion analysis” to “policy analysis”.

Reviewer 3 Report
A very well organised paper, my opinion is that it must be published without any modificationsAuthor Response
Responses to Review 3
Thank you very much for your helpful feedback and insightful comments. We have taken all of the suggestions and comments into consideration during this revision. We truly appreciate the opportunity to revise our paper, and believe that our manuscript has significantly improved in response to the ideas and recommendations of the review team.
Detailed responses:
A very well organised paper, my opinion is that it must be published without any modifications.
Our Response: We greatly appreciate your detailed and specific comments, which is our greatest recognition.

Reviewer 4 Report
Dear authors
Let me congratulate you for the soundness of this manuscript and its clarity. The introduction provides a great context to the problem under investigation. The literature is well construed and articulated. The methods used are clear and sound. The results are well explained and articulated with the literature review. The only aspect that needs strengthening is the conclusion as it reads as rushed and it could provide an overview on the limitations of this study and what could potentially be addressed in the future.
Author Response
Responses to Review 4
Thank you very much for your helpful feedback and insightful comments. We have taken all of the suggestions and comments into consideration during this revision. We truly appreciate the opportunity to revise our paper, and believe that our manuscript has significantly improved in response to the ideas and recommendations of the review team.
Detailed responses:
Let me congratulate you for the soundness of this manuscript and its clarity. The introduction provides a great context to the problem under investigation. The literature is well construed and articulated. The methods used are clear and sound. The results are well explained and articulated with the literature review. The only aspect that needs strengthening is the conclusion as it reads as rushed and it could provide an overview on the limitations of this study and what could potentially be addressed in the future.
Our Response: Thank you for pointing this out. We have outlined in 5.1 the limitations of this study and what could potentially be addressed in the future. And We have also increased the content of the conclusion to make it more substantial. Details are as follows.
5.1. Conclusion
This research uses the dynamic two-stage directional distance function DEA model, with environmental exogenous variables, to measure the mining production-land rehabilitation efficiencies of 29 provinces in China and arrived at the following conclusions.
The overall efficiencies of most provinces in China are in general below 0.5. Among them, the efficiencies of Tianjin, Inner Mongolia, and Heilongjiang reached the DEA optimal level, which closely relate to the distribution of local natural resources and policy guidance. The efficiencies of Anhui, Chongqing, Jiangsu, and Liaoning were relatively low. There was a big fluctuation in the efficiency of the land rehabilitation stage in the four years, but the average efficiency was slightly higher than that in the mining production stage. Thus, the fluctuation of land restoration efficiency should be reduced to make it grow steadily better. For the process of demand and supply transformation of the mining industry, China still faces greater risks and problems that must be improved.
By comparing the efficiencies of the mining production stage and the land rehabilitation stage, we find that the efficiency of the former is relatively low. Thus, mining efficiency should urgently be improved in order to promote the overall efficiency of a region.
The efficiency distribution of China’s provinces is not uniform, which relates to regional policy and resource distribution. For overall efficiency levels, Jiangsu, Anhui, Shaanxi, and Chongqing are lower, while Inner Mongolia, Heilongjiang, and Tianjin are higher.
This paper employs theoretical analysis of 29 provinces’ mining efficiency in China. The aim is to effectively guide each province in mining and rehabilitation and to deal with the lack of specialization in China for this sector presently and in future. While this research paper’s sample selection is scientific and objective, it fails to fully simulate the various emergencies of mining and land restoration.
5.2. Policy recommendations
Due to the great difference between the resource distribution and efficiencies of provinces in China, the projects for improving the efficiency level of each province are different. Thus, corresponding policies need to strengthen these efficiencies. Based on the development efficiency for the period 2014-17, the following suggestions are made.
5.2.1. Integrate mining resources
Through the market guidance mechanism, developed provinces should be able to control their mining output and promote the sustainable development of mining areas. Speeding up the business of mining area integration can help to gradually replace the situation of decentralized mining of small- and medium-sized mines in China. Integrating regional resources can spur a region to achieve a good state of economies of scale and scope. By integrating resources, mining areas with similar resource reserves and quality can share technology, machinery, and equipment and thus reduce repeated investment in fixed assets. This is also conducive to the spillover effect in the land rehabilitation stage, so that mining areas with a similar environment and damage conditions can reduce rehabilitation investment and obtain better rehabilitation of land area. China should continue to promote the improvement of efficiency in the land rehabilitation stage. Provinces with low efficiency levels like Shaanxi can learn from provinces with a similar climate environment and mining area types, introduce advanced mining and land rehabilitation technology locally, and implement technology improvement so as to increase their level of efficiency. The government should strengthen with the surrounding water, circuit and other infrastructure. On the premise of not affecting the life of residents around the area, the government can give priority to providing cheap hydropower resources and mine road subsidies for mining areas. Integration of surrounding infrastructure can achieve the purpose of resource integration around a mining area.
5.2.2. Eliminate backward capacity and technology
Mining firms can phase out backward production capacity and replace old existing fixed assets by introducing high-tech mining machines and accessories. At the same time, they can set up corresponding research bases according to the unique situation locally, increase R&D investment, strengthen their scientific and technological levels, change from rough mining to scientific mining, and conform to the trend of high-quality development in China. In addition, a special group for land rehabilitation should set up a scientific and technological research base. By analyzing the local mining environment and soil conditions, damaged or occupied land can be restored into a historical mining park or reclaimed as cultivated land, in order to give full play to the residual value of damaged soil. Such actions can strengthen the efficiency of the land rehabilitation stage.
In view of the amount of seriously damaged land, biochemical measures should be introduced in each area to gradually repair damaged land and improve the environment of mining areas by building a stable ecosystem. For the mountainous provinces of Guizhou, Sichuan, and Chongqing, environmental restoration should be planned according to the geological conditions of mines in each area. For the land restoration of mining areas in mountainous regions, crops with economic value and in line with China’s market, such as artemisia argyi or Conyza canadensis (L.) Cronq., can be selected as vegetation restoration options in southwest part of the country. Depending on varying slopes, different vegetation can be adopted to firm up the slopes and prevent the occurrence of landslides, so as to ensure smooth mining and land restoration. For winter freezing conditions in areas like Xinjiang, Heilongjiang, Jilin, Liaoning, and Inner Mongolia, it is necessary to solve the problem of frozen soil. By introducing new winter mining technology, the impact of bad weather on mining should be able to slow down. For the current operation of mining machines, it is possible to retrieve the braking and driving devices of crushers or excavators and clean their internal widgets after each use, especially those that are easily damaged by freezing. Mining firms must pay attention to the temperature while cleaning and check whether the widgets are leaking.
5.2.3. Strengthen government administrative measures
For serious damage caused by mining, local governments should intervene in administrative management and guide the coordinated development of mining production and land rehabilitation. National and local people's congressional bodies should enhance the laws and regulations on the examination and approval of mining licenses, the handling of safety production licenses, and the division of responsibility for mining accidents and further guide the coordinated development of regional production and environment. As to the mining sector, each province can give appropriate tax exemptions to mining enterprises that run under high efficiency and strong land rehabilitation efficiency according to the situation of their own conditions. Through the leading role of excellent mining enterprises, the efficiency of land rehabilitation can be improved. For some old mining areas that are short of funds, each province should set up special cash reserves that can support old mining areas to update their own equipment and strengthen their efficiency level. In order to improve the use of special funds for the environmental restoration of mining areas in each province, local governments should closely track the whereabouts of the funds and present restoration results in real time, thus curbing any misappropriation of money or financial corruption. Implementing these measures should help China to clean up the land damage caused by mining in local mining areas and promote the improvement of overall efficiency in this important industrial sector.
